# PARETO LOW-RANK ADAPTERS: EFFICIENT MULTI-TASK LEARNING WITH PREFERENCES

**Nikolaos Dimitriadis**
EPFL

**Pascal Frossard**
EPFL

**François Fleuret**
University of Geneva, Meta FAIR

## ABSTRACT

Multi-task trade-offs in machine learning can be addressed via Pareto Front Learning (PFL) methods that parameterize the Pareto Front (PF) with a single model. PFL permits to select the desired operational point during inference, contrary to traditional Multi-Task Learning (MTL) that optimizes for a single trade-off decided prior to training. However, recent PFL methodologies suffer from limited scalability, slow convergence, and excessive memory requirements, while exhibiting inconsistent mappings from preference to objective space. We introduce PaLoRA, a novel parameter-efficient method that addresses these limitations in two ways. First, we augment any neural network architecture with task-specific low-rank adapters and continuously parameterize the PF in their convex hull. Our approach steers the original model and the adapters towards learning general and task-specific features, respectively. Second, we propose a deterministic sampling schedule of preference vectors that reinforces this division of labor, enabling faster convergence and strengthening the validity of the mapping from preference to objective space throughout training. Our experiments show that PaLoRA outperforms state-of-the-art MTL and PFL baselines across various datasets, scales to large networks, reducing the memory overhead $23.8 - 31.7$ times compared with competing PFL baselines in scene understanding benchmarks.

## 1 INTRODUCTION

Building machine learning models with multi-task capabilities is becoming increasingly prevalent (Ruder, 2017; Crawshaw, 2020), pursuing the advantages of a shared representation coupled with the practical benefits of a single model, in terms of memory requirements and inference times. However, the construction of generalist agents (Reed et al., 2022) by solving simultaneously multiple tasks introduces conflicts and there often does not exist a single optimal solution. Multi-objective optimization (MOO) problems rather have a set of optimal solutions, formally known as the *Pareto Front* (PF), each corresponding to a different trade-off among the objectives.

Multi-Task Learning (MTL) algorithms optimize for a single trade-off, delivering one point in the PF (Mahapatra & Rajan, 2020; Lin et al., 2019a; Cipolla et al., 2018; Chen et al., 2018; Sener & Koltun, 2018; Navon et al., 2022), but lack the flexibility of dynamic adaptation to unseen preferences during inference. In recommender systems, for instance, objectives such as semantic relevance and revenue or content quality introduce trade-offs (Lin et al., 2019b). Similar competing objectives can arise in autonomous self-driving (Wang et al., 2018), multi-objective image or protein generation (Yao et al., 2024), and aligning LLMs to multiple preferences (Zhong et al., 2024). *Pareto Front Learning* (PFL) methodologies (Navon et al., 2021; Ruchte & Grabocka, 2021; Dimitriadis et al., 2023) address the inherent scalability issues of the MTL discrete solution set by directly parameterizing the entire PF. They learn a conditional model, where providing the desired preference as input generates the weights of a neural network satisfying that trade-off.

Despite these advancements, the practical application of PFL methods remains constrained by several challenges, including high memory consumption and unreliable preference-to-objective

---

Contact: `nikolaos.dimitriadis@epfl.ch`. Code available at github.com/nik-dim/palora.

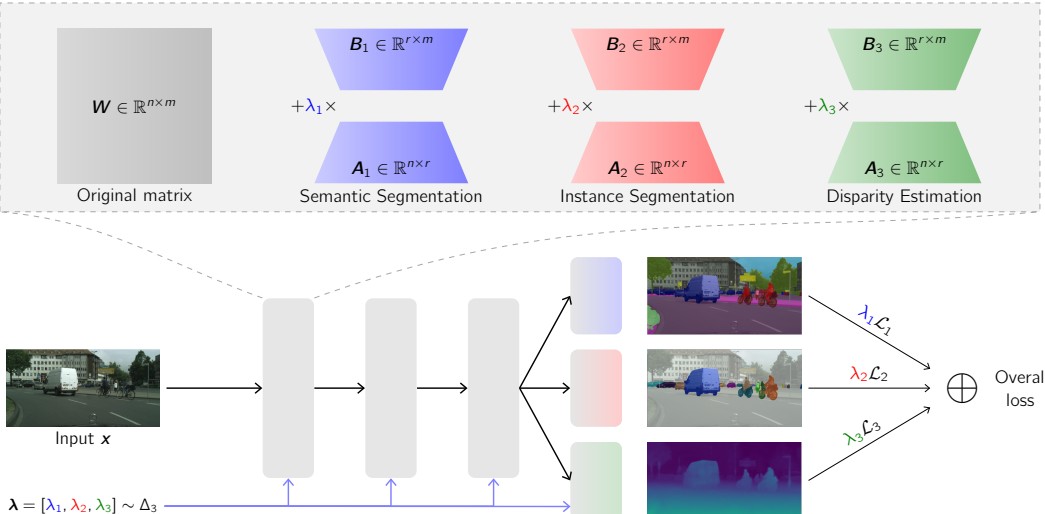

Figure 1: **Conceptual illustration of the architecture**. Each layer consists of the base network's weight matrix $W$ and 3 low-rank adapters $\{(A_t, B_t)\}_{t=1}^3$. During training, we sample preference $\lambda = [\lambda_1, \lambda_2, \lambda_3] \sim \Delta_3$, each layer's weights are formed by the weighted sum of the original matrix and the tasks' low-rank adapters. The overall loss uses the *same* $\lambda$ to weigh the task losses, steering each adapter to learn task-specific features and the shared backbone to learn a general representation.

mappings. Specifically, current methods rely on training a conditional model by sampling preferences from a Dirichlet distribution, and using these preferences to weigh the objective losses. While this approach theoretically enables coverage of the entire PF, it introduces significant inefficiencies for PFL approaches compared to MTL ones. First, the weight-generating mechanism often requires parameters far exceeding those of the target network, leading to substantial memory overheads (Navon et al., 2021; Dimitriadis et al., 2023). Second, the reliance on extensive training periods and complex learning rate schedules results in slower convergence (Navon et al., 2021; Ruchte & Grabocka, 2021). Finally, the inherent randomness in preference sampling reduces control over the optimization process and compounds on these issues. The stochasticity can lead to inconsistent mappings, where adjusting the preference towards a particular task may not yield improved performance in that task. Overall, these limitations hinder both the efficiency and the reliability of PFL models.

In this paper, we introduce P̲areto L̲ow-R̲ank A̲daptors (PaLoRA), a novel PFL algorithm that tackles these limitations through two complementary strategies. First, we equip neural networks with task-specific low-rank adapters (Hu et al., 2022) and establish a *Pareto Set* within their convex hull, as in Figure 1. The core network parameters learn a general representation beneficial for all objectives, whereas the adapters acquire specialized features. Second, we propose to replace the random preference sampling with a deterministic procedure, presented in Figure 2, that reinforces this division of labor. Initially, the focus lies on building general features and towards the latest stages of training on specialization, by selecting preferences in the middle and edges of the simplex, respectively.

PaLoRA significantly reduces memory overhead by a structured division of labor within the model, while the core neural network parameters remain fixed, responsible for learning shared representations, only the participation of the lightweight low-rank adapters varies. The deterministic sampling schedule mirrors the natural learning process observed in neural networks: early training phases prioritize learning general features, followed by specialization to task-specific features—a strategy akin to curriculum learning (Soviany et al., 2022). By focusing on the center of the preference simplex during early training and progressively exploring its edges, PaLoRA achieves a functionally diverse PF without sacrificing convergence speed. Finally, PaLoRA is inspired by pareameter-efficient fine-tuning methods (Hu et al., 2022) and, therefore, inherently supports *Pareto expansion* (Ma et al., 2020), i.e., enabling the method to serve as a fine-tuning mechanism for extending the PF in the vicinity of pre-trained solutions. This opens up opportunities to leverage the representations learned by foundation models, making PaLoRA highly adaptable and scalable across tasks.

Our experimental results demonstrate that PaLoRA significantly outperforms other PFL methods across multiple benchmarks using very small ranks ($r \leq 4$), has better Pareto alignment and requires fewer parameters. For instance, when training occurs from scratch for `CityScapes`, PaLoRA requires a $4.2\%$ memory overhead compared to MTL baselines, a $23.8\times$ overhead reduction compared to PaMaL (Dimitriadis et al., 2023), while showing superior performance. When applied as a fine-tuning mechanism, the proposed method is able to quickly align the adapters towards continuously modeling the PF locally, offering a flexible Pareto expansion mechanism.

Our contributions are as follows:

- We present PaLoRA, a novel PFL method that augments any architecture with task-specific per layer low-rank adapters and discovers a Pareto Set in the their convex hull.
- PaLoRA uses annealed *deterministic* preference sampling to guide PFL training, and we show that it improves on training stability and the PF functional diversity.
- Our experiments on multiple benchmarks and across model sizes show that PaLoRA outperforms PFL baselines, achieving higher hypervolume while incurring orders of magnitude less memory overhead. For `CityScapes` and `NYUv2`, PaLoRA introduces an overhead of $4.2\%$ and $6.3\%$, respectively, marking a $23.8\times$ and $31.7\times$ reduction compared PaMaL (Dimitriadis et al., 2023), while staying competitive or outperforming MTL baselines.
- We show that the proposed method can be easily adapted to a fine-tuning mechanism to expand the PF continuously in the neighborhood of a pretrained checkpoint.

## 2 RELATED WORK

**Paremeter-Efficient Fine-Tuning** (PEFT) strategies (Lester et al., 2021; Li & Liang, 2021; Houlsby et al., 2019) have advanced the adaptation of pre-trained models to specific tasks without extensive re-training. Fine-tuning pre-trained models occurs in low-dimensional subspaces (Aghajanyan et al., 2021), and Hu et al. (2022) proposed fine-tuning only a low-rank decomposition of each layer's weight matrix, reducing computational and memory costs and sparking a series of subsequent methodologies (Huang et al., 2023; Valipour et al., 2023; Kopiczko et al., 2024; Hyeon-Woo et al., 2022; Yeh et al., 2023; Liu et al., 2024b; Tian et al., 2024; Wu et al., 2024). Low rank adapters have also been studied from a multi-task perspective, Feng et al. (2024) use a routing mechanism after fine-tuning several LoRAs, while Zhao et al. (2024) adaptively retrieve and compose multiple LoRAs depending on the input prompts. Compared to these works, we employ low-rank adapters throughout training and not just as a fine-tuning mechanism and towards approximating the PF instead of optimizing over a single static objective.

**Multi-Task Learning.** The pursuit of learning multiple tasks within a single model has deep roots in machine learning (Caruana, 1997; Argyriou et al., 2008; Ruder, 2017; Crawshaw, 2020), evolving in the deep learning era through architectural innovations (Misra et al., 2016; Ma et al., 2018; Ruder et al., 2019) and optimization techniques (Cipolla et al., 2018; Chen et al., 2018; Yu et al., 2020; Liu et al., 2020). The methodologies focus on combining the task contributions, either directly on the loss level (Cipolla et al., 2018; Lin et al., 2022), or on the gradient level (Chen et al., 2018; Liu et al., 2022; 2021; Chen et al., 2020). However, these methods only produce a single point in the Pareto Front and do not offer users control during inference.

**Pareto Front Learning** methods parameterize the Pareto Front, allowing the construction of a model satisfying a user's trade-off at inference and at no cost. Sener & Koltun (2018) scale Multiple Gradient Descent Algorithm (Désidéri, 2012) to deep learning settings. Lin et al. (2019a) introduce constraints encoding trade-offs to steer solutions along the Pareto Front (Fliege & Svaiter, 2000), but require separate training runs per solution. Navon et al. (2021); Lin et al. (2021); Hoang et al. (2023) use HyperNetworks (Ha et al., 2017) to *continuously* approximate the Pareto Front, but even with chunking to reduce memory, they still double the GPU memory requirements due to simultaneous deployment with the target network. Ruchte & Grabocka (2021) offer a memory-efficient approach via input augmentation, though network conditioning issues remain. Dimitriadis et al. (2023) use weight ensembles but require a copy of the model for each task, leading to excessive memory overhead. In contrast, our approach maintains a stable core network for general features while aligning task-specific adapters to improve convergence. We finally note that a concurrent work (Chen & Kwok, 2024) also employed LoRAs (Hu et al., 2022) towards approximating the Pareto Front. Beyond the architectural component, we also propose a deterministic preference schedule to improve on the validity of the final Pareto Front and convergence.

---

**Algorithm 1:** PaLoRA

---

**Input:** neural network $f$, multi-task loss function $\mathcal{L}$, training dataset $\mathcal{D}_{train}$

**6** **for** *batch* $(\boldsymbol{x}, \boldsymbol{y}) \in \mathcal{D}_{train}$ **do**

**1 Function** `Preference(`$\tau, M, Q$`)`:

**7**     $\mathcal{L}_{total} \leftarrow 0$

**2**     Distribute evenly in the $T$-dimensional simplex $M$ points $\boldsymbol{\lambda}^{(1)}, \ldots, \boldsymbol{\lambda}^{(M)}$

**8**     $\boldsymbol{\lambda}^{(1)}, \ldots, \boldsymbol{\lambda}^{(M)} \leftarrow$ `Preference(`$\tau, M, Q$`)`

**9**     **for** $\boldsymbol{\lambda} \in \left\{ \boldsymbol{\lambda}^{(1)}, \ldots, \boldsymbol{\lambda}^{(M)} \right\}$ **do**

**3**     Anneal the rays with temperature $Q$ via Equation 3 for timestep $\tau$

     // form parameters via Equation 1

**10**      $\boldsymbol{\theta} \leftarrow \left\{ \boldsymbol{W}^{(\ell)} + \frac{\alpha}{r} \sum_{t=1}^{T} \lambda_t \boldsymbol{A}_t^{(\ell)} \boldsymbol{B}_t^{(\ell)} \right\}_{\ell=1}^{L}$

**4**     **return** $\boldsymbol{\lambda}^{(1)}, \ldots, \boldsymbol{\lambda}^{(M)}$

**11**      $\mathcal{L}_{total} \leftarrow \mathcal{L}_{total} + \boldsymbol{\lambda}^\top \mathcal{L}\left( f(\boldsymbol{x}; \boldsymbol{\theta}), \boldsymbol{y} \right)$

    // $\tau$ parameterizes the ray preference distribution

**12**     Update $\boldsymbol{\theta}_W, \boldsymbol{\theta}_A, \boldsymbol{\theta}_B$ via backpropagation

**5** $\tau \leftarrow 0$

**13**     $\tau \leftarrow \tau + 1$

---

## 3   MULTI-OBJECTIVE OPTIMIZATION AND PARETO FRONT LEARNING

Consider a dataset $\mathcal{D} = \{(\boldsymbol{x}^{(i)}, \boldsymbol{y}^{(i)})\}_{i=1}^{N}$ for samples $\boldsymbol{x}^{(i)} \in \mathcal{X}$ and labels $\boldsymbol{y}^{(i)} \in \mathcal{Y}$ and $T$ tasks. Each task $t \in [T]$ is associated with a loss $\mathcal{L}_t : \mathcal{Y}_t \times \mathcal{Y}_t \to \mathbb{R}_+$. The overall goal corresponds to the multi-objective optimization problem of minimizing the vector loss $\boldsymbol{L} = [\mathcal{L}_1, \ldots, \mathcal{L}_T]$.

We assume a general multi-task architecture comprising of an encoder $g(\boldsymbol{x}, \boldsymbol{\theta}_{enc}) : \mathcal{X} \times \Theta_{enc} \to \mathcal{Z}$, mapping to embedding space $\mathcal{Z}$ and multiple decoders $f_t(\boldsymbol{z}, \boldsymbol{\theta}_t) : \mathcal{Z} \times \Theta_t \to \mathcal{Y}_t, t \in [T]$, where $\theta_{(\cdot)}$ represents network parameters. Multi-Task Learning focuses on learning a model $f : \mathcal{X} \times \Theta \to \mathcal{Y}$ that performs well on all tasks. However, in the setting of vector optimization (Boyd & Vandenberghe, 2004), no single solution $\boldsymbol{\theta}$ can be optimal for all tasks simultaneously. In contrast, optimality is sought in the Pareto sense, i.e., a solution $\boldsymbol{\theta}^*$ is Pareto optimal if there is no other solution $\boldsymbol{\theta}'$ that is better for all tasks. Formally:

**Definition 1** (Pareto Optimality). Let $\mathcal{L}_t(\mathcal{Y}_t, f(\mathcal{X}_t, \boldsymbol{\theta})) = \mathcal{L}_t(\boldsymbol{\theta})$. Assume two solutions $\boldsymbol{\theta}, \boldsymbol{\theta}' \in \Theta$ in parameter space. A point $\boldsymbol{\theta}$ dominates a point $\boldsymbol{\theta}'$ if $\mathcal{L}_t(\boldsymbol{\theta}) \leq \mathcal{L}_t(\boldsymbol{\theta}') \; \forall t \in [T]$ and $\boldsymbol{L}(\boldsymbol{\theta}) \neq \boldsymbol{L}(\boldsymbol{\theta}')$. Then, a point $\boldsymbol{\theta}$ is called Pareto optimal if there exists no $\boldsymbol{\theta}'$ that dominates it. The set of Pareto optimal points forms the *Pareto Set* $\mathcal{P}_S$ and its image in objective space is known as the *Pareto Front* $\mathcal{P}_F$.

Let $\Delta_T = \{\boldsymbol{\lambda} \in \mathbb{R}_+^T : \sum_{t=1}^T \lambda_t = 1\}$ be the $T$-dimensional simplex representing the set of all possible user preferences and $\ell : \Theta \to \mathbb{R}_+^T$ a mapping from weight to objective space, i.e., the vector loss $\ell(\boldsymbol{\theta}) = [\mathcal{L}_1(\boldsymbol{\theta}), \ldots, \mathcal{L}_T(\boldsymbol{\theta})]$. For a given *preference vector* $\boldsymbol{\lambda} \in \Delta_T$ as input, the overall objective of PFL is a weight generating mechanism $h : \Delta_T \to \Theta$ such that $\mathcal{P}_S = h(\Delta_T)$ and the function $\ell \circ h$ is monotonic, i.e., increasing the importance of one task leads to its loss decrease. Our goal is discovering a *Pareto Subspace* (Dimitriadis et al., 2023), i.e., a low-dimensional subspace that approximates the PF in a single training run.

## 4   PALORA: PARETO LOW-RANK ADAPTERS

PFL methodologies have two components: the conditional model $h$ mapping from preference space to the Pareto Set and the distribution used during training to sample preferences. We improve on both of these axes in Section 4.1 and Section 4.2, respectively, and present them jointly in algorithm 1.

### 4.1   WEIGHT GENERATING MECHANISM OF PALORA

Consider the case of a single layer parameterized by $\boldsymbol{W} \in \mathbb{R}^{n \times m}$. We omit bias terms for simplicity. The output of the layer is $\boldsymbol{y} = \boldsymbol{W}\boldsymbol{x}$ for $\boldsymbol{x} \in \mathbb{R}^m$. Inspired by the weight ensembling approach of PaMaL (Dimitriadis et al., 2023), we propose that each task $t \in [T]$ augments the set of learnable parameters by introducing a copy of the model:

$$\boldsymbol{y}' = \left( \boldsymbol{W} + \sum_{t=1}^T \lambda_t \boldsymbol{W}_t \right) \boldsymbol{x} \tag{1}$$

for preference $\boldsymbol{\lambda}$ in the $T$-dimensional simplex, i.e., $\boldsymbol{\lambda} = [\lambda_1, \ldots, \lambda_T]^\top \in \Delta_T$. Given the similarity among ensemble members (Dimitriadis et al., 2023), the full-rank duplicates can be replaced with

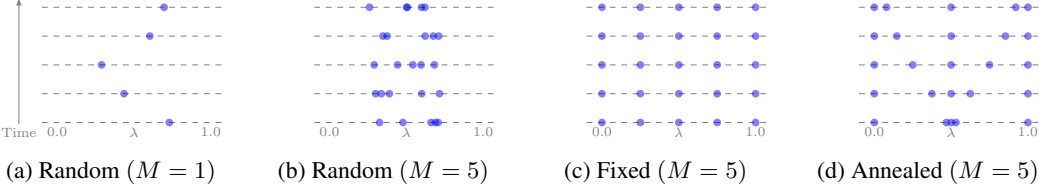

(a) Random $(M = 1)$     (b) Random $(M = 5)$     (c) Fixed $(M = 5)$     (d) Annealed $(M = 5)$

Figure 2: **Random vs deterministic preference schedules** for two tasks as a function of time. Each dashed line corresponds to a different batch, bottom is beginning of training and top end of training. For each batch, preferences $\boldsymbol{\lambda} = [\lambda, 1-\lambda]$ are drawn; we only show $\lambda$. Randomly sampling (a) $M = 1$ ray per batch or (b) multiple $(M > 1)$ rays (Dimitriadis et al., 2023) can lead to poor mappings from preference to objective space due to lack of exploration and tightly clustered sampled rays. Instead, our (c) proposed deterministic schedule resolves these issues and (d) our temperature annealing, focusing progressively more to learning task-specific features, can lead to wider Pareto Fronts.

low-rank matrices $\boldsymbol{A}_t \in \mathbb{R}^{n \times r}, \boldsymbol{B}_t \in \mathbb{R}^{r \times m}, t \in [T]$ and $r \ll \min\{n, m\}$:

$$\boldsymbol{W}_t = \boldsymbol{A}_t \boldsymbol{B}_t \implies \boldsymbol{y}' = \left(\boldsymbol{W} + \frac{\alpha}{r} \sum_{t=1}^{T} \lambda_t \boldsymbol{A}_t \boldsymbol{B}_t \right) \boldsymbol{x} \tag{2}$$

where $\alpha$ a scaling factor that tunes the emphasis on the low-rank residuals (Huh et al., 2024). Besides the above LoRA (Hu et al., 2022) formulation, the task-specific weights $\boldsymbol{W}_t$ can be constructed by any approaches in the low-rank family (Yeh et al., 2023; Zhang et al., 2023; Kopiczko et al., 2024).

During each training iteration, preferences $\boldsymbol{\lambda}$ are drawn, and a direct link between each task and its corresponding adapter is formed by defining the total loss as $\boldsymbol{\lambda}^\top \boldsymbol{L} = \sum_{t=1}^{T} \lambda_t \mathcal{L}_t$. This alignment ensures that the contribution of each adapter matches the loss weight of the corresponding task. Figure 1 illustrates the method in the case of three tasks. PaLoRA establishes a clear division of labor between the base network and the task-specific adapters. The former's involvement in the weight generation mechanism of Equation 2 is not a function of the drawn preference $\boldsymbol{\lambda}$ but always present, helping convergence by keeping the major part of the representation steady. Following the Dimitriadis et al. (2023, Theorem 4.2), Theoreom 2 shows that PaLoRA can approximate the PF due to the universal approximation theorem (Cybenko, 1989). The proof is provided in the appendix.

**Theorem 2.** *Let* $f_t : \mathcal{X} \times \Theta \mapsto \mathcal{Y}$ *be a family of continuous mappings, where* $t = 1, \ldots, T$, *and* $\mathcal{X} \subset \mathbb{R}^D$ *is compact. Then,* $\forall \epsilon > 0$, *there exists a ReLU multi-layer perceptron* $f$ *with three different weight parameterizations* $\boldsymbol{\theta}_0, \boldsymbol{\theta}_1, \boldsymbol{\theta}_2 \in \Theta$, *such that* $\forall t \in [T], \exists \alpha \in [0, 1], \forall \boldsymbol{x} \in \mathcal{X}$:

$$|f_t(\boldsymbol{x}) - f(\boldsymbol{x}; \boldsymbol{\theta}_0 + \alpha \boldsymbol{\theta}_1 + (1 - \alpha) \boldsymbol{\theta}_2)| \le \epsilon.$$

PaLoRA is based on parameter-efficient approaches (Lester et al., 2021; Houlsby et al., 2019; Mahabadi et al., 2021; Ben Zaken et al., 2022; Hu et al., 2022) and is, therefore, by design compatible with pre-trained models and able to take advantage of the knowledge embedded in foundation models (Bommasani et al., 2021). In contrast, previous PFL solutions, e.g., hypernetworks, are not easily adaptable to such settings and do not scale to the size of such large models. As a consequence, our method can be launched as a second phase of training, similar to the setting studied by Ma et al. (2020), where once a multi-task solution has been attained, e.g., using NashMTL (Navon et al., 2022) or linear scalarization, a second phase expands the PF in the neighborhood of the solution.

**Memory Complexity** For a linear layer with $mn$ parameters, the PaLoRA layer requires $mn + Tr(m + n)$. Modulating the rank $r$ controls the layer's expressivity and number of parameters, allowing for task-specific features to be steered to the main backbone or the adapters.

## 4.2    IMPROVED CONVERGENCE AND FUNCTIONAL DIVERSITY WITH DETERMINISTIC PREFERENCE SCHEDULE

Pareto Front Learning methods lead to an increase of trainable parameters and lower convergence rates compared to MTL approaches. The preference distribution dictates the construction of the models during training, e.g., via Equation 2 in the case of PaLoRA or as an input to a hypernetwork (Navon et al., 2021) or as mixing coefficients of a weight ensemble (Dimitriadis et al., 2023).

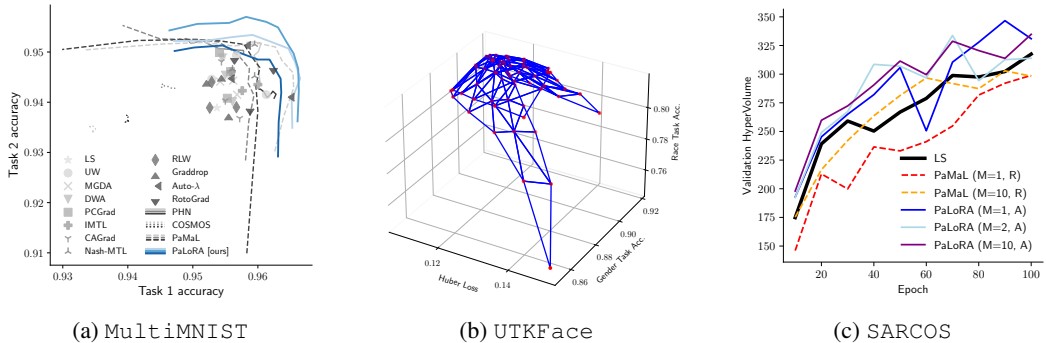

Figure 3: **Experimental results**. (a) PaLoRA outperforms MTL baselines and achieves higher Hypervolume while requiring less memory vs other PFL algorithms, (b) constructs a wide Pareto Front for benchmarks with two classification and one regression tasks. (c) Even for 7 tasks, PaLoRA showcases fast convergence while PaMaL is slow due to $7\times$ increase in parameter count.

However, previous works assume the distribution to be (uniform) Dirichlet and suffer from slow convergence rates and inconsistent mappings from preference to objective space. Consider the case of drawing a single random weighting $\boldsymbol{\lambda}$ on every batch as in Figure 2a. In the case of two tasks, for instance, where the ray is $(\lambda, 1-\lambda)$ for $\lambda \in [0, 1]$, two consecutive rays can be $\boldsymbol{\lambda}_1 = (1-\epsilon_1, \epsilon_1)$ and, $\boldsymbol{\lambda}_2 = (1 - \epsilon_2, \epsilon_2)$ or $\boldsymbol{\lambda}_2' = (\epsilon_2', 1 - \epsilon_2')$ for $\epsilon_1, \epsilon_2, \epsilon_2' \approx 0$. In the former case ($\boldsymbol{\lambda}_1 \to \boldsymbol{\lambda}_2$), starvation occurs for the second task and the representation drifts towards accommodating the first, while in the latter case ($\boldsymbol{\lambda}_1 \to \boldsymbol{\lambda}_2'$) the representation abruptly changes leading to instability. The lack of control introduces variance in the gradient updates, since $\boldsymbol{\lambda}$ determines both the model's representation and the overall scalarized objective. Performing multiple forward passes (Dimitriadis et al., 2023) with different rays, as in Figure 2b, can decrease the variance of the updates but the lack of exploration can still lead to invalid mappings from preference space to weight space and, finally, objective space. Specifically, sampled rays from a tightly clustered region of the probability space result in similar weight configurations, e.g. via Equation 2, and given the non-convexity of the loss landscape their small differences may not correspond to vector loss objectives that satisfy the Pareto properties defined in Definition 1. Overall, stochasticity in the updates can lead to decrease in loss but may violate the premise of Pareto Front Learning in creating a valid mapping from preference to objective space.

We therefore propose to replace random sampling with a deterministic schedule. Assuming a budget of $M$ forward passes and $T$ tasks, we sample preferences $\{\tilde{\boldsymbol{\lambda}}_m\}_{m=1}^M$ *evenly spaced* in the simplex $\Delta_T$, as in Figure 2c. Given that neural networks initially learn general features and specialize during the later training stages (Zeiler & Fergus, 2014; Kalimeris et al., 2019; Valeriani et al., 2023), we propose to sample multiple preference vectors per batch with a schedule mimicking the training dynamics. For time $\tau \in [0, 1]$, where $\tau = 0$ and $\tau = 1$ correspond to the beginning and end of training, respectively, we anneal the preferences $\{\tilde{\boldsymbol{\lambda}}_m\}_{m=1}^M$ as follows:

$$\boldsymbol{\lambda}_m(\tau) = g_{\tau,Q}(\tilde{\boldsymbol{\lambda}}) := \frac{\tilde{\boldsymbol{\lambda}}_m^{\tau'}}{\sum_{t=1}^T \tilde{\lambda}_{m,t}^{\tau'}} \quad \text{for } \tau' = \frac{\tau}{Q} \tag{3}$$

where $Q$ controls the temperature of the annealing. In the initial training stages, the preferences are concentrated in the center and progressively more to the faces of the simplex, as shown in Figure 2d for 2 dimensions and in Figure 11 for 3 dimensions. The schedule is deterministic in order to minimize the effect of pernicious updates due to potential imbalanced samplings and therefore we do not consider regularization terms (Dimitriadis et al., 2023; Ruchte & Grabocka, 2021), also eliminating their associated hyperparameters.

## 5 EXPERIMENTS

We evaluate PaLoRA on a variety of tasks and datasets, ranging from multi-label classification to complex scene understanding benchmarks in CityScapes and NYUv2. We include datasets ranging from 2 to 7 tasks and architectures from LeNet to SegNet (Badrinarayanan et al., 2017).

Table 1: `CityScapes`: Test performance averaged over 3 seeds, $\Delta_p$ is the parameter count increase w.r.t. the MTL model. We highlight the **best** and second best results per task. PaLoRA outperforms PFL and MTL methods, while slightly increasing parameter count and allowing for user control.

| | | Segmentation | | Depth | | $\Delta_p\%\downarrow$ | Controllable |
|---|---|---|---|---|---|---|---|
| | | mIoU ↑ | Pix Acc ↑ | Abs Err ↓ | Rel Err ↓ | | |
| | STL | 70.96 | 92.12 | 0.0141 | 38.6435 | 0% | ✗ |
| MTL | LS (Caruana, 1997) | 70.12 | 91.90 | 0.0192 | 124.0615 | 0% | ✗ |
| | UW (Cipolla et al., 2018) | 70.20 | 91.93 | 0.0189 | 125.9433 | 0% | ✗ |
| | MGDA (Sener & Koltun, 2018) | 66.45 | 90.79 | 0.0141 | 53.1376 | 0% | ✗ |
| | DWA (Liu et al., 2019) | 70.10 | 91.89 | 0.0192 | 127.6589 | 0% | ✗ |
| | PCGrad (Yu et al., 2020) | 70.02 | 91.84 | 0.0188 | 126.2551 | 0% | ✗ |
| | IMTL (Liu et al., 2020) | 70.77 | 92.12 | 0.0151 | 74.2300 | 0% | ✗ |
| | CAGrad (Liu et al., 2021) | 69.23 | 91.61 | 0.0168 | 110.1387 | 0% | ✗ |
| | Nash-MTL (Navon et al., 2022) | **71.13** | **92.23** | 0.0157 | 78.4993 | 0% | ✗ |
| | RLW (Lin et al., 2022) | 68.79 | 91.52 | 0.0213 | 126.9423 | 0% | ✗ |
| | Graddrop (Chen et al., 2020) | 70.07 | 91.93 | 0.0189 | 127.1464 | 0% | ✗ |
| | RotoGrad (Javaloy & Valera, 2022) | 69.92 | 91.85 | 0.0193 | 127.2806 | 0% | ✗ |
| | Auto-$\lambda$ (Liu et al., 2022) | 70.47 | 92.01 | 0.0177 | 116.9594 | 0% | ✗ |
| PFL | COSMOS (Ruchte & Grabocka, 2021) | 69.78 | 91.79 | 0.0539 | 136.614 | ≪ 1% | ✓ |
| | PaMaL (Dimitriadis et al., 2023) | 70.35 | 91.99 | 0.0141 | 54.520 | 100% | ✓ |
| | PaLoRA [ours] | 71.11 | 92.21 | **0.0140** | **51.2672** | 4.2% | ✓ |

**Baselines.** We compare against a diverse set of algorithms, including Single-Task Learning (STL), MTL methodologies based on loss balancing (Caruana, 1997; Cipolla et al., 2018; Liu et al., 2019; Lin et al., 2022) and gradient balancing (Sener & Koltun, 2018; Yu et al., 2020; Liu et al., 2020; Chen et al., 2020; Liu et al., 2021; Navon et al., 2022; Liu et al., 2022; Javaloy & Valera, 2022). We consider PFL baselines in Pareto HyperNetwork (Navon et al., 2021, PHN), COSMOS (Ruchte & Grabocka, 2021) and Pareto Manifold Learning (Dimitriadis et al., 2023, PaMaL). For PFL methods, evaluation is performed on a grid of $K$ evenly spaced points spanning the $T$-dimensional simplex.

## 5.1 MULTI-LABEL CLASSIFICATION

First, we test the effectiveness of PaLoRA on `MultiMNIST`, a digit classification dataset based on `MNIST` and use a LeNet architecture (Lecun et al., 1998), trained for 10 epochs, and present the results in Figure 3a. Compared to standard MTL approaches that result in a single point in objective space, PFL methodologies discover a subspace of solutions, parameterized by the user preference $\boldsymbol{\lambda}$ in Equation 2. We use rank $r = 1$, resulting in 12.5% additional trainable parameters compared to the base multi-task model, i.e., the scatter points in the plot. In contrast, PaMaL (Dimitriadis et al., 2023) doubles the trainable parameters, PHN (Navon et al., 2021) requires $\times 100$ memory overhead and COSMOS (Ruchte & Grabocka, 2021) has low hypervolume. Overall, our method achieves superior performance compared to other PFL methods, while requiring far less parameters, and showing that even a rank of $r = 1$ suffices to cover the Pareto Front.

## 5.2 SCALING UP THE NUMBER OF TASKS

We explore benchmarks beyond two tasks and present the results in Figure 3 qualitatively for 3 tasks and in Figure 3c quantitatively for 7 tasks. We consider `UTKFace` (Zhang et al., 2017), a dataset with images and three tasks of gender and ethnicity classification and age regression, and we use a ResNet (He et al., 2016) backbone following prior work (Dimitriadis et al., 2023). We observe in Figure 3b that our proposed method is effective at discovering a valid Pareto Front. Similarly, we include in the appendix results on `MultiMNIST-3` (Dimitriadis et al., 2023), a generalization of the previous dataset with three objectives, where PaLoRA is able to discover a wide and functionally diverse Pareto Front. Finally, we explore `SARCOS` (Vijayakumar & Schaal, 2000), a robotic dataset with $T = 7$ regression tasks that predict joint torques based on joint positions, velocities, and accelerations. Figure 3c shows that PaLoRA converges faster than Linear Scalarization, a MTL method used as control, and considerably faster compared to state-of-the-art PaMaL (Dimitriadis et al., 2023), due to the latter requiring the joint optimization of $T = 7$ copies of the model. In contrast, we use rank $r = 4$, leading to a parameter increase of 62%. Overall, PaLoRA scales to a larger number of tasks compared to PaMaL, showcasing superior performance and lower memory requirements.

Table 2: $\texttt{NYUv2}$: Test performance averaged over 3 seeds. $\Delta_p$ is the parameter count increase w.r.t. the multi-task model. We highlight the **best** and second best results per task. PaLoRA is superior to PaMaL (Dimitriadis et al., 2023) while requiring $31.7\times$ less parameters.

| | | Segmentation | | Depth | | Surface Normal | | | | | $\Delta_p\%\downarrow$ | Controllable |
| | | | | | | Angle Distance ↓ | | Within $t°$ ↑ | | | | |
| | | mIoU ↑ | Pix Acc ↑ | Abs Err ↓ | Rel Err ↓ | Mean | Median | 11.25 | 22.5 | 30 | | |
|---|---|---|---|---|---|---|---|---|---|---|---|---|
| | STL | 36.58 | 62.62 | 0.6958 | 0.2737 | 25.27 | 18.25 | 32.33 | 59.09 | 70.00 | 0% | ✗ |
| MTL | LS (Caruana, 1997) | 36.33 | 62.98 | 0.5507 | 0.2224 | 27.71 | 21.84 | 26.44 | 51.72 | 63.90 | 0% | ✗ |
| | UW (Cipolla et al., 2018) | 31.68 | 60.11 | 0.5509 | 0.2242 | 28.30 | 22.59 | 25.74 | 50.33 | 62.49 | 0% | ✗ |
| | MGDA (Sener & Koltun, 2018) | 31.19 | 60.22 | 0.5745 | 0.2267 | **24.84** | **18.30** | **32.45** | **59.04** | **70.28** | 0% | ✗ |
| | DWA (Liu et al., 2019) | 37.09 | 63.69 | 0.5463 | 0.2225 | 27.57 | 21.79 | 26.55 | 51.78 | 64.02 | 0% | ✗ |
| | PCGrad (Yu et al., 2020) | 36.83 | 63.43 | 0.5504 | 0.2182 | 27.44 | 21.57 | 26.90 | 52.25 | 64.40 | 0% | ✗ |
| | IMTL (Liu et al., 2020) | 36.43 | 63.96 | 0.5437 | 0.2203 | 25.87 | 19.63 | 30.02 | 56.19 | 67.95 | 0% | ✗ |
| | Nash-MTL (Navon et al., 2022) | 37.66 | 64.75 | **0.5279** | **0.2109** | 25.56 | 19.30 | 30.59 | 56.90 | 68.55 | 0% | ✗ |
| | RLW (Lin et al., 2022) | 33.86 | 61.49 | 0.5692 | 0.2271 | 29.06 | 23.68 | 24.06 | 48.27 | 60.67 | 0% | ✗ |
| | Graddrop (Chen et al., 2020) | 36.98 | 63.31 | 0.5423 | 0.2204 | 27.64 | 21.89 | 26.38 | 51.64 | 63.93 | 0% | ✗ |
| PFL | PaMaL (Dimitriadis et al., 2023) | 33.94 | 62.55 | 0.5592 | 0.2188 | 26.60 | 20.33 | 29.09 | 54.61 | 66.35 | 200% | ✓ |
| | PaLoRA [ours] | **38.27** | **64.79** | 0.5370 | 0.2150 | 25.66 | 19.34 | 30.47 | 56.90 | 68.56 | 6.3% | ✓ |

## 5.3 Scene Understanding

We now evaluate PaLoRA on large scale datasets of scene understanding. An example of input and label combinations is given in Figure 1. Our experimental setup is based on previous MTL works (Liu et al., 2019; Yu et al., 2020; Liu et al., 2021; Navon et al., 2022; Dimitriadis et al., 2023).

**CityScapes** (Cordts et al., 2016) contains high-resolution urban street images and we focus on the tasks of semantic segmentation and depth regression. We train a SegNet (Badrinarayanan et al., 2017) for 100 epochs, using Adam optimizer (Kingma & Ba, 2015) with learning rate $10^{-4}$ that is halved after 75 epochs. The results are presented in Table 1 for rank $r = 4$ and $M = 5$. In terms of PFL baselines, while COSMOS (Ruchte & Grabocka, 2021) increases slightly the number of trainable parameters, its poor performance on the task of depth estimation renders it non-competitive. We note that we do not consider PHN (Navon et al., 2021), since it requires the ad hoc definition of a hypernetwork architecture with chunking for the weight generating mechanism, introducing many hyperparameters. Additionally, the hypernetwork requires at least as many parameters as the target network to match its expressiveness, effectively doubling the memory requirements. For this reason, Navon et al. (2021) only consider ENet (Paszke et al., 2016), a network of only 0.37M parameters compared to $> 25M$ of the SegNet architecture. Compared to PaMaL (Dimitriadis et al., 2023), our method leads to improvements across tasks while reducing the memory overhead $\sim 23.8$ times.

**NYUv2** Similar to previous MTL works, we consider the $\texttt{NYUv2}$ dataset (Silberman et al., 2012) for the tasks of semantic segmentation, depth estimation, and surface normal prediction, and report the results in Table 2. We reserve 95 images for validation and use a setup similar to $\texttt{CityScapes}$ but train for 200 epochs. The full experimental details are provided in the appendix. To the best of our knowledge, and due to scalability, PaLoRA is the first PFL method to explore benchmarks as challenging as $\texttt{NYUv2}$. Additionally to PHN (Navon et al., 2021), we omit the COSMOS baseline (Ruchte & Grabocka, 2021) due to its poor performance on $\texttt{CityScapes}$. PaLoRA scales to the complexity of the benchmark without incurring large memory costs, while PaMaL (Dimitriadis et al., 2023) requires 3 times the memory of the original model, PaLoRA needs 6.3% more parameters.

## 5.4 Continuous Pareto Expansion

While previous sections focused on training models from scratch, PaLoRA can also be used as a second-stage fine-tuning approach, similar to the original use of low-rank adapters (Hu et al., 2022). Following Ma et al. (2020), we expand the Pareto Front locally around a pre-trained model $\theta_0$, using checkpoints trained with linear scalarization. Only adapters are fine-tuned, matching the final stage's learning rate, with training lasting 4 epochs for $\texttt{MultiMNIST}$ and 5 epochs for $\texttt{CityScapes}$, consuming just 40% and 5% of the original training budgets. Results in Figure 4 show PaLoRA effectively expands the Pareto Front, enhancing initial model performance. The scaling factor $\alpha$ controls the spread, while the middle plot of Figure 4 shows that the annealed deterministic sampling gradually increases functional diversity. Unlike the costly checkpoint

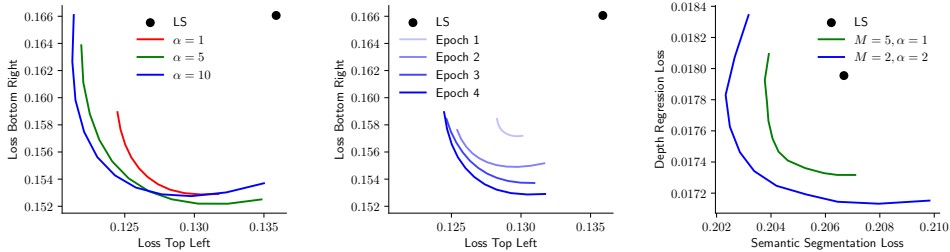

Figure 4: **Pareto Front Expansion**. Given a checkpoint $\theta_0$, marked as •, PaLoRA expands locally the Pareto Front in its neighborhood $\mathcal{N}(\theta_0)$. (Left) The scaling $\alpha$ of Equation 2 determines the functional diversity of the final `MultiMNIST` Front. (Middle) The epoch-by-epoch progression of the `MultiMNIST` Pareto Front expansion. (Right) The final `CityScapes` Pareto Front.

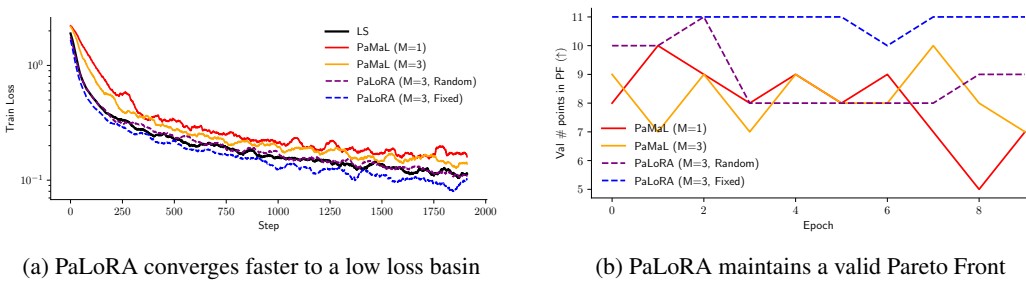

(a) PaLoRA converges faster to a low loss basin    (b) PaLoRA maintains a valid Pareto Front

Figure 5: PaLoRA satisfies both PFL goals: high controllability coupled with superior performance. PaLoRA converges faster than state-of-the-art PFL method in PaMaL and on par or better to MTL methods. It is also more consistent in terms of the validity of the Pareto Front across epochs, while the number of points in the Pareto Front in PaMaL varies a lot.

storage of Ma et al. (2020), PaLoRA incurs only a $4.2\%$ memory overhead, forming a continuous linear segment in weight space that maps to the Pareto Front.

# 6 DISCUSSION

## 6.1 DETERMINISTIC SCHEDULE IMPROVES CONVERGENCE AND PARETO DIVERSITY

We evaluate the effect of the sampling procedure as a function of the number of samples $M \in \mathbb{N}$, whether they are drawn from a random distribution or deterministically and the effect of annealed schedule. Specifically, we consider the following distributions over preference rays:

$$
\mathbf{\Lambda}_\tau = \left[ \boldsymbol{\lambda}_\tau^{(1)}, \ldots, \boldsymbol{\lambda}_\tau^{(M)} \right] \text{ for } \boldsymbol{\lambda}_\tau^{(m)} \sim
\begin{cases}
\text{Dirichlet}(p\mathbf{1}), & \text{Random, } M = 1 \text{ (Navon et al., 2021)} \\
\text{Dirichlet}(p\mathbf{1}), & \text{Random, } M > 1 \text{ (Dimitriadis et al., 2023)} \\
g_{1,Q}(\tilde{\boldsymbol{\lambda}}_m), & \text{Deterministic [\textbf{ours}]} \\
g_{\tau,Q}(\tilde{\boldsymbol{\lambda}}_m), & \text{Deterministic Annealed [\textbf{ours}]}
\end{cases}
$$

As $\tau$ increases the sampled weightings shift focus from the center of the simplex towards the edges. We consider the setting outlined in Section 5.1 for the `MultiMNIST` dataset and compare our proposed method with PaMaL (Dimitriadis et al., 2023) and Linear Scalarization (LS), a Multi-Task Learning method used as control. We present the results in Figure 5 in terms of two metrics; the progression of the training loss captures how fast each method reaches *a neighborhood* of low loss, while the number of points[*] in the validation Pareto Front measures the degree of *Pareto alignment* of said neighborhood. We observe that PaLoRA outperforms PaMaL in both metrics. Specifically, PaMaL has slower convergence since it has to simultaneously optimize two copies of the model, while the low-rank adapters of our proposed method are lightweight and efficient. PaLoRA also has faster convergence compared to a single-point algorithm in LS, since sampling multiple rays

---

[*]We sample 11 points in total $\{(\lambda, 1 - \lambda) \text{ for } \lambda = 0, 0.1, \ldots, 0.9, 1\}$.

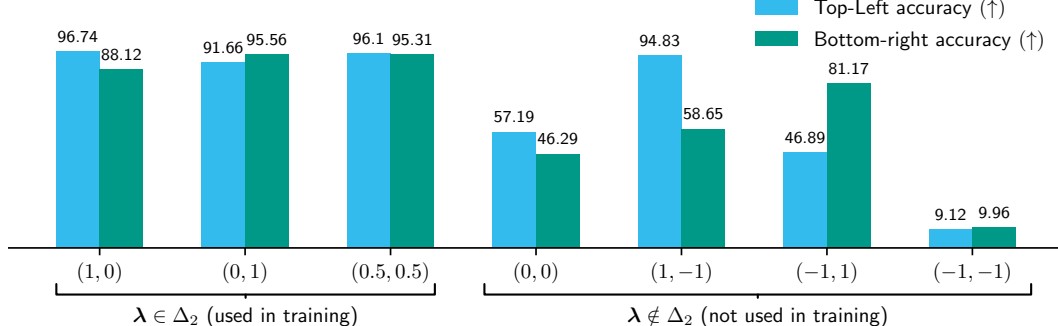

Figure 6: **Adapters contain task-specific information.** While selecting rays $\boldsymbol{\lambda} \in \Delta_2$ used in training maintains high performance on both tasks, *pseudopreferences* $\boldsymbol{\lambda} \notin \Delta_2$ reveal that adapters are task-specific: negating the contribution of one leads to forgetting the associated task.

acts as a regularization effect (Izmailov et al., 2018; Foret et al., 2021). More importantly, the deterministic schedule of the proposed method maintains a valid Pareto Front throughout training, as indicated in Figure 5b. In contrast, both PaMaL and low rank adapters without the proposed deterministic schedule fluctuate in the number of validation points in the Pareto Front, highlighting that stochasticity undermines the control desideratum of *Pareto Front Learning*.

We also conduct a larger scale search on the effectiveness of deterministic sampling in constructing valid Pareto Fronts in Appendix D where we consider the cases of $M \in \{3, 5, 7, 9\}$, deterministic and Dirichlet sampling, fixed or annealed schedule and several temperatures for each sampling category and compare the HyperVolume (Zitzler & Thiele, 1999), which measures solutions quality, and the number of points in the Pareto Front. The results indicate that random sampling introduces instability: while it can achieve high HyperVolume, it often leads to a dominated Pareto Front. In contrast, deterministic sampling offers a well-distributed Pareto Front, is more robust to hyperparameter variations, and enables effective performance with lower $M$, reducing memory overhead from multiple forward passes.

## 6.2 DIVISION OF LABOR BETWEEN CORE NEURAL NETWORK AND ADAPTERS

We investigate if the low-rank adapters contain task-specific information. While the rays used during training lie in simplex $\Delta_T$, we examine the impact of escaping the simplex to highlight the separation of feature building objectives among the adapters. We evaluate *pseudopreferences* $\boldsymbol{\lambda} \in \{(0, 0), (1, -1), (-1, 1)\} \not\subset \Delta_2$, corresponding to models with general features, and those forgetting the second and first tasks, respectively. For comparison, we also evaluated actual preferences $\boldsymbol{\lambda} \in \{(1, 0), (0, 1), (0.5, 0.5)\} \subset \Delta_2$, with results shown in Figure 6 for the MultiMNIST benchmark. The findings reveal the specialized roles of adapters: removing the first task's contribution in $\boldsymbol{\lambda} = (-1, 1)$ notably impairs its performance, while the second task is less affected. The core network without adapter contributions for $\boldsymbol{\lambda} = (0, 0)$ lacks discriminative features, and $\boldsymbol{\lambda} = (-1, -1)$ results in random predictions. This aligns with task arithmetic evaluations (Ilharco et al., 2023; Ortiz-Jimenez et al., 2023; Yadav et al., 2023; Wang et al., 2024), where negating task contributions leads to forgetting. However, unlike independently trained models, our jointly trained adapters still allow knowledge transfer, avoiding a complete separation of task-specific knowledge.

## 7 CONCLUSION AND LIMITATIONS

In conclusion, PaLoRA addresses the limitations of existing PFL methodologies by introducing a parameter-efficient approach that augments the original model with task-specific low-rank adapters coupled with a deterministic preference sampling to guide PFL training. Our proposed method effectively separates the learning of general and task-specific features, facilitated by a carefully designed sampling schedule. Our extensive experimental results demonstrate that PaLoRA not only surpasses PFL baselines across benchmarks but also achieves scalability to larger networks with limited memory overhead. By providing a continuous parameterization of the Pareto Front and ensuring efficient use of memory, PaLoRA offers a promising solution for real-world applications.

ACKNOWLEDGMENTS

We would like to thank Guillermo Ortiz-Jiménez, Alessandro Favero, Ke Wang, Ortal Senouf and the anonymous reviewers for their valuable feedback.

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

# A ADDITIONAL RESULTS

## A.1 SARCOS

We further explore the setting for SARCOS dataset, presented in Section 5.2. The experimental setting remains the same except the number of epochs, which have been increased from 100 to 500. The results are presented in Figure 7 and show that PaLoRA converges faster to a higher HyperVolume compared to PaMaL (Dimitriadis et al., 2023).

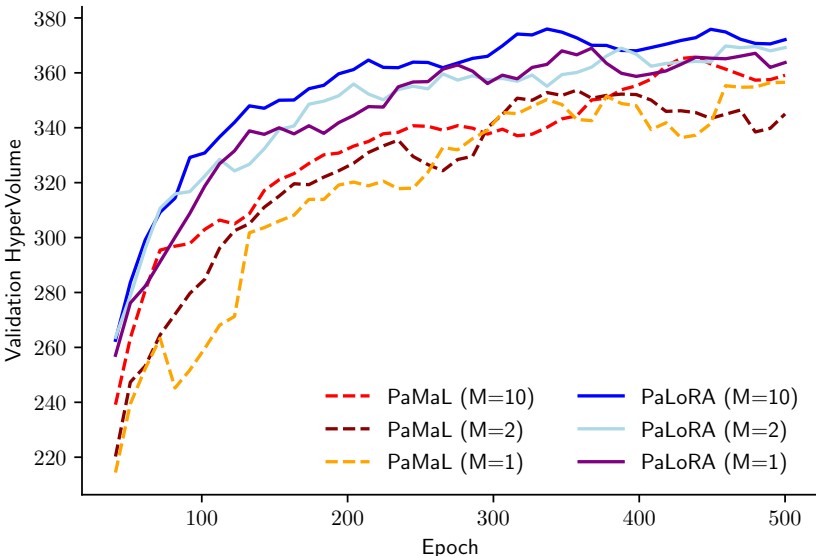

Figure 7: Comparison between PaMaL and PaLoRA on the SARCOS dataset for longer training times.

## A.2 ABLATION ON RANK $r$

Table 3 presents an ablation on the LoRA rank $r$ for CityScapes. Specifically, we consider ranks $r \in \{1, 2, 4, 8\}$. We also explore the axis of the number of forward passes $M$. We use 3 seeds per combination and report the mean.

Table 3: Ablation on rank $r$ for CityScapes.

| $r$ | $M$ | Segmentation | | Depth | |
|---|---|---|---|---|---|
| | | mIOU ↑ | Pix Acc ↑ | Abs Err ↓ | Rel Err ↓ |
| 1 | 3 | 71.56 | 92.35 | 0.0139 | 52.4928 |
| 1 | 5 | 71.55 | 92.32 | 0.0139 | 48.7060 |
| 2 | 3 | 71.44 | 92.37 | 0.0144 | 45.5715 |
| 2 | 5 | 71.47 | 92.26 | 0.0139 | 49.1715 |
| 4 | 3 | 71.58 | 92.37 | 0.0141 | 43.2215 |
| 4 | 5 | 71.44 | 92.29 | 0.0141 | 41.2195 |
| 8 | 3 | 71.44 | 92.41 | 0.0144 | 49.9970 |
| 8 | 5 | 71.64 | 92.38 | 0.0147 | 48.9209 |

## A.3 DETAILED RESULTS FOR MULTIMNIST AND NYUV2

Table 4 and Table 5 present the results for the MultiMNIST and NYUv2 benchmarks, respectively, including standard deviations. Three seeds are used for each method in both benchmarks.

Table 4: Detailed results for `MultiMNIST`

|          | Top-Left | Bottom-Right |
|----------|----------|--------------|
| LS       | $95.47_{\pm 0.08}$ | $94.45_{\pm 0.43}$ |
| UW       | $95.70_{\pm 0.34}$ | $94.51_{\pm 0.32}$ |
| MGDA     | $95.57_{\pm 0.11}$ | $94.33_{\pm 0.13}$ |
| DWA      | $95.52_{\pm 0.07}$ | $94.48_{\pm 0.36}$ |
| PCGrad   | $95.51_{\pm 0.07}$ | $94.56_{\pm 0.37}$ |
| IMTL     | $95.78_{\pm 0.17}$ | $94.40_{\pm 0.16}$ |
| CAGrad   | $95.55_{\pm 0.12}$ | $94.17_{\pm 0.47}$ |
| NashMTL  | $95.84_{\pm 0.16}$ | $94.78_{\pm 0.27}$ |
| RLW      | $95.41_{\pm 0.19}$ | $94.06_{\pm 0.24}$ |
| GradDrop | $95.40_{\pm 0.14}$ | $94.24_{\pm 0.34}$ |
| AutoL    | $95.94_{\pm 0.39}$ | $94.57_{\pm 0.43}$ |
| RotoGrad | $95.92_{\pm 0.25}$ | $94.48_{\pm 0.44}$ |
| PHN      | $96.04_{\pm 0.20}$ | $94.91_{\pm 0.46}$ |
| COSMOS   | $94.08_{\pm 0.50}$ | $93.90_{\pm 0.35}$ |
| PAMAL    | $96.17_{\pm 0.27}$ | $95.32_{\pm 0.15}$ |
| PaLoRA   | $96.55_{\pm 0.13}$ | $95.39_{\pm 0.24}$ |

Table 5: Detailed results for `NYUv2`.

|     |          | Segmentation | | Depth | | Surface Normal | | | | |
|-----|----------|--------------|---------|--------------|--------------|----------------|---------|---------|---------|---------|
|     |          | mIoU ↑ | Pix Acc ↑ | Abs Err ↓ | Rel Err ↓ | Angle Distance ↓ | | Within $t°$ ↑ | | |
|     |          |        |           |           |           | Mean | Median | 11.25 | 22.5 | 30 |
|     | STL      | $36.58_{\pm 0.57}$ | $62.62_{\pm 0.11}$ | $0.6958_{\pm 0.0402}$ | $0.2737_{\pm 0.0039}$ | $25.27_{\pm 0.17}$ | $18.25_{\pm 0.33}$ | $32.33_{\pm 0.52}$ | $59.09_{\pm 0.67}$ | $70.0_{\pm 0.51}$ |
| MTL | LS       | $36.33_{\pm 2.43}$ | $62.98_{\pm 0.86}$ | $0.5507_{\pm 0.0055}$ | $0.2224_{\pm 0.0102}$ | $27.71_{\pm 0.73}$ | $21.84_{\pm 0.86}$ | $26.44_{\pm 1.25}$ | $51.72_{\pm 1.68}$ | $63.9_{\pm 1.58}$ |
|     | UW       | $31.68_{\pm 3.37}$ | $60.11_{\pm 2.07}$ | $0.5509_{\pm 0.0225}$ | $0.2242_{\pm 0.0092}$ | $28.3_{\pm 0.82}$ | $22.59_{\pm 1.09}$ | $25.74_{\pm 1.5}$ | $50.33_{\pm 2.07}$ | $62.49_{\pm 1.92}$ |
|     | MGDA     | $31.19_{\pm 0.76}$ | $60.22_{\pm 1.06}$ | $0.5745_{\pm 0.0303}$ | $0.2267_{\pm 0.0078}$ | $24.84_{\pm 0.37}$ | $18.3_{\pm 0.51}$ | $32.45_{\pm 1.18}$ | $59.04_{\pm 1.03}$ | $70.28_{\pm 0.74}$ |
|     | DWA      | $37.09_{\pm 2.83}$ | $63.69_{\pm 1.88}$ | $0.5463_{\pm 0.0093}$ | $0.2225_{\pm 0.0105}$ | $27.57_{\pm 0.65}$ | $21.79_{\pm 0.76}$ | $26.55_{\pm 1.1}$ | $51.78_{\pm 1.49}$ | $64.02_{\pm 1.39}$ |
|     | PCGrad   | $36.83_{\pm 1.36}$ | $63.43_{\pm 0.76}$ | $0.5504_{\pm 0.0009}$ | $0.2182_{\pm 0.0036}$ | $27.44_{\pm 0.41}$ | $21.57_{\pm 0.43}$ | $26.9_{\pm 0.73}$ | $52.25_{\pm 0.87}$ | $64.4_{\pm 0.81}$ |
|     | IMTL     | $36.43_{\pm 2.03}$ | $63.96_{\pm 0.89}$ | $0.5437_{\pm 0.0196}$ | $0.2203_{\pm 0.0053}$ | $25.87_{\pm 0.28}$ | $19.63_{\pm 0.3}$ | $30.02_{\pm 0.64}$ | $56.19_{\pm 0.67}$ | $67.95_{\pm 0.59}$ |
|     | NashMTL  | $37.66_{\pm 2.02}$ | $64.75_{\pm 0.85}$ | $0.5279_{\pm 0.0131}$ | $0.2109_{\pm 0.0037}$ | $25.56_{\pm 0.26}$ | $19.3_{\pm 0.3}$ | $30.59_{\pm 0.71}$ | $56.9_{\pm 0.62}$ | $68.55_{\pm 0.52}$ |
|     | RLW      | $33.86_{\pm 0.67}$ | $61.49_{\pm 1.5}$ | $0.5692_{\pm 0.0159}$ | $0.2271_{\pm 0.0085}$ | $29.06_{\pm 0.54}$ | $23.68_{\pm 0.59}$ | $24.06_{\pm 0.65}$ | $48.27_{\pm 1.02}$ | $60.67_{\pm 1.19}$ |
|     | GradDrop | $36.98_{\pm 1.92}$ | $63.31_{\pm 1.14}$ | $0.5423_{\pm 0.0125}$ | $0.2204_{\pm 0.008}$ | $27.64_{\pm 0.71}$ | $21.89_{\pm 0.74}$ | $26.38_{\pm 0.95}$ | $51.64_{\pm 1.49}$ | $63.93_{\pm 1.46}$ |
| PFL | PaMaL    | $33.94_{\pm 1.29}$ | $62.55_{\pm 0.85}$ | $0.5592_{\pm 0.0035}$ | $0.2188_{\pm 0.0024}$ | $26.6_{\pm 0.13}$ | $20.33_{\pm 0.21}$ | $29.09_{\pm 0.24}$ | $54.61_{\pm 0.47}$ | $66.34_{\pm 0.45}$ |
|     | PaLoRA   | $38.27_{\pm 0.8}$ | $64.79_{\pm 0.55}$ | $0.537_{\pm 0.0065}$ | $0.215_{\pm 0.0047}$ | $25.66_{\pm 0.18}$ | $19.34_{\pm 0.23}$ | $30.47_{\pm 0.19}$ | $56.9_{\pm 0.43}$ | $68.56_{\pm 0.42}$ |

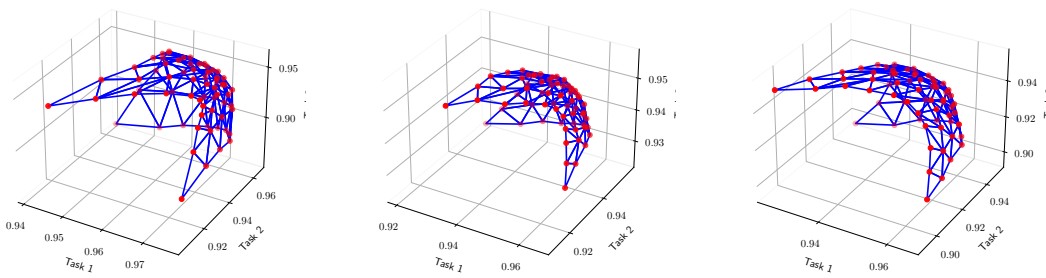

Figure 8: Qualitative results on `MultiMNIST-3`, measuring accuracy on three digit classification tasks. PaLoRA is able to cover the entire Pareto Front.

# B EXPERIMENTAL DETAILS

All experiments are conducted with `PyTorch`(Paszke et al., 2019) in Tesla V100-SXM2-32GB GPUs. We use three seeds per method. Our source code extends the codebases of previous works (Dimitriadis et al., 2023; Navon et al., 2022; Liu et al., 2019). We use the `MultiMNIST` variant from Dimitriadis et al. (2023). For `CityScapes` and `NYUv2`, we use the variants from Liu et al. (2019), which are also explored in Navon et al. (2022); Liu et al. (2021; 2024a; 2022).

## B.1 FULL TRAINING

**MultiMNIST**  We use the same settings as Dimitriadis et al. (2023). For all experiments the rank is set to $r = 1$. We perform a grid search on the scaling $\alpha \in \{1, 2, 5, 10\}$ and annealing temperature $T$ and report the results in Figure 9 and Figure 10 in terms of validation accuracy and loss. We use 3 seeds per configuration and report the results that achieve the highest average hypervolume (Zitzler & Thiele, 1999). Training lasts 10 epochs.

**MultiMNIST-3**  The experimental settings are the same as `MultiMNIST`, apart from the number of epochs that has been increased from 10 to 20.

**UTKFace**  We use the same settings as Dimitriadis et al. (2023).

**SARCOS**  We follow the experimental protocol presented by Navon et al. (2021).

**CityScapes**  We use the same settings as (Dimitriadis et al., 2023). A modified version of the same experiment was also presented in (Liu et al., 2019; Yu et al., 2020; Liu et al., 2021; Navon et al., 2022); the difference is the existence of a validation set and training lasts 100 epochs. We use the SegNet architecture (Badrinarayanan et al., 2017), not the MTAN (Liu et al., 2019) variant for computational reasons. We train for 100 epochs.

**NYUv2**  We use the experimental settings from Navon et al. (2022). We use a SegNet architecture (Badrinarayanan et al., 2017), not the MTAN (Liu et al., 2019) variant for computational reasons. We use 95 out of 795 training images for validation. We train for 200 epochs.

## B.2 RUNTIME

The runtime depends on the number of forward passes, denoted as $M$. We also perform evaluation on a held-out dataset every $k$ epochs. Evaluation for PFL is expensive since we evaluate the performance for models formed for various preferences $\boldsymbol{\lambda} \in \Delta_T$. Hence, evaluation time scales linearly with the number of sampled points. For instance, for two tasks, we use 11 evenly spaced points in [0,1]. For $M = 1$, the experiments on `MultiMNIST`, `CityScapes`, `NYUv2` take $\sim 1$ min, $\sim 90$ mins and $\sim 4.5$ hours. For the scene understanding benchmarks, we use a gradient balancing algorithm similar to (Chen et al., 2018) and used in (Dimitriadis et al., 2023), since the results were superior. Gradient balancing algorithms have longer runtimes compared to loss-balancing ones, e.g., (Caruana, 1997; Cipolla et al., 2018). The above runtime numbers refer to gradient balancing.

## B.3 PARETO EXPANSION

We use exactly the same settings as in full training, the only difference is the number of epochs. For `MultiMNIST` we use 4 epochs and for `CityScapes` 5 epochs. In both cases, the initial checkpoint, denoted as $\boldsymbol{\theta}_0$ in the main text, corresponds to the first seed of the Linear Scalarization (Caruana, 1997) baseline.

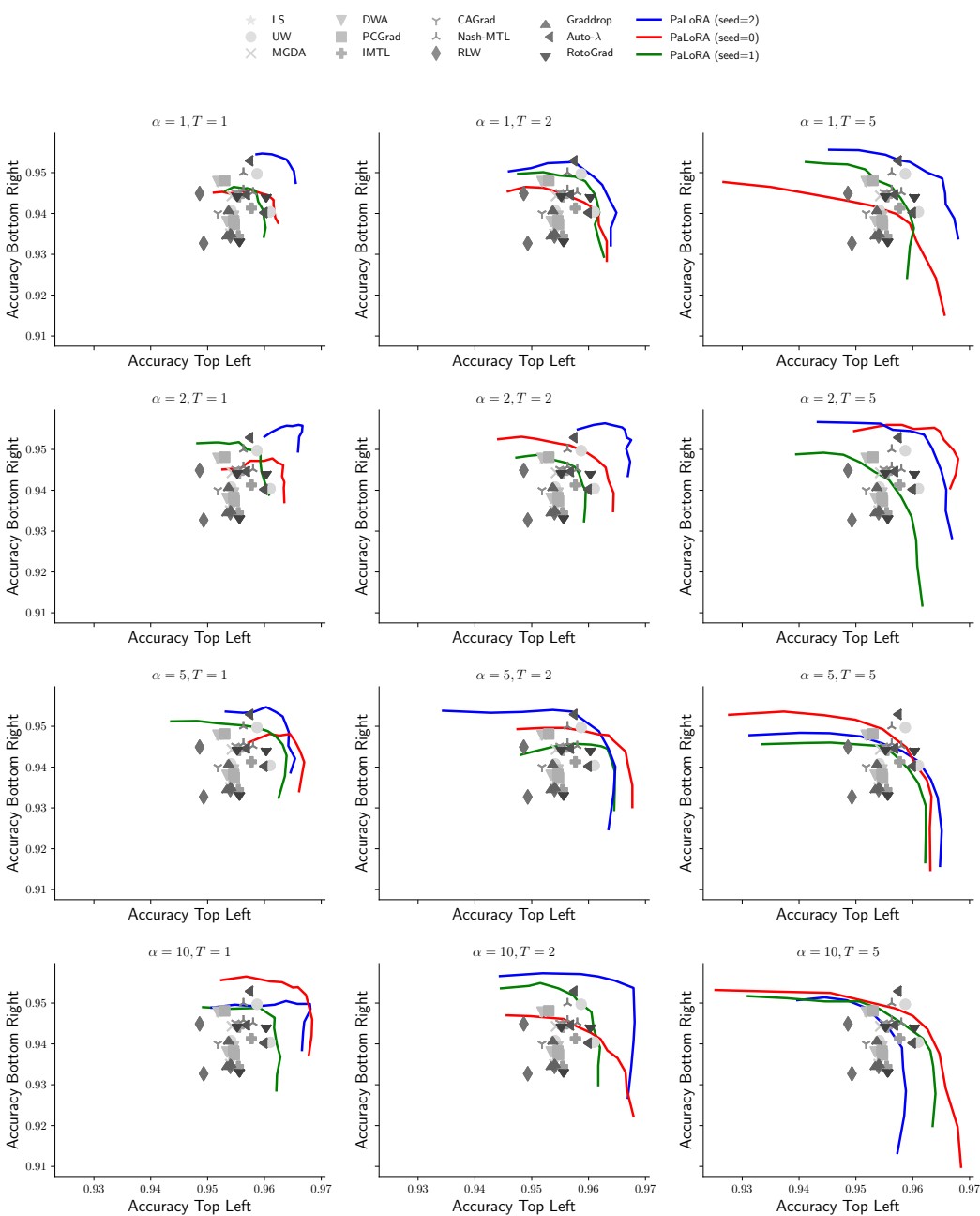

Figure 9: Ablation on the scaling parameter $\alpha$, defined in Equation 2, and the annealing temperature $T$ for the validation set of MultiMNIST. We omit PFL baselines to reduce visual clutter and present 3 seeds for each configuration. We observe that higher temperatures and scales leads to larger Pareto Fronts. Figure 3a presents the test results for the configuration with the highest mean hypervolume.

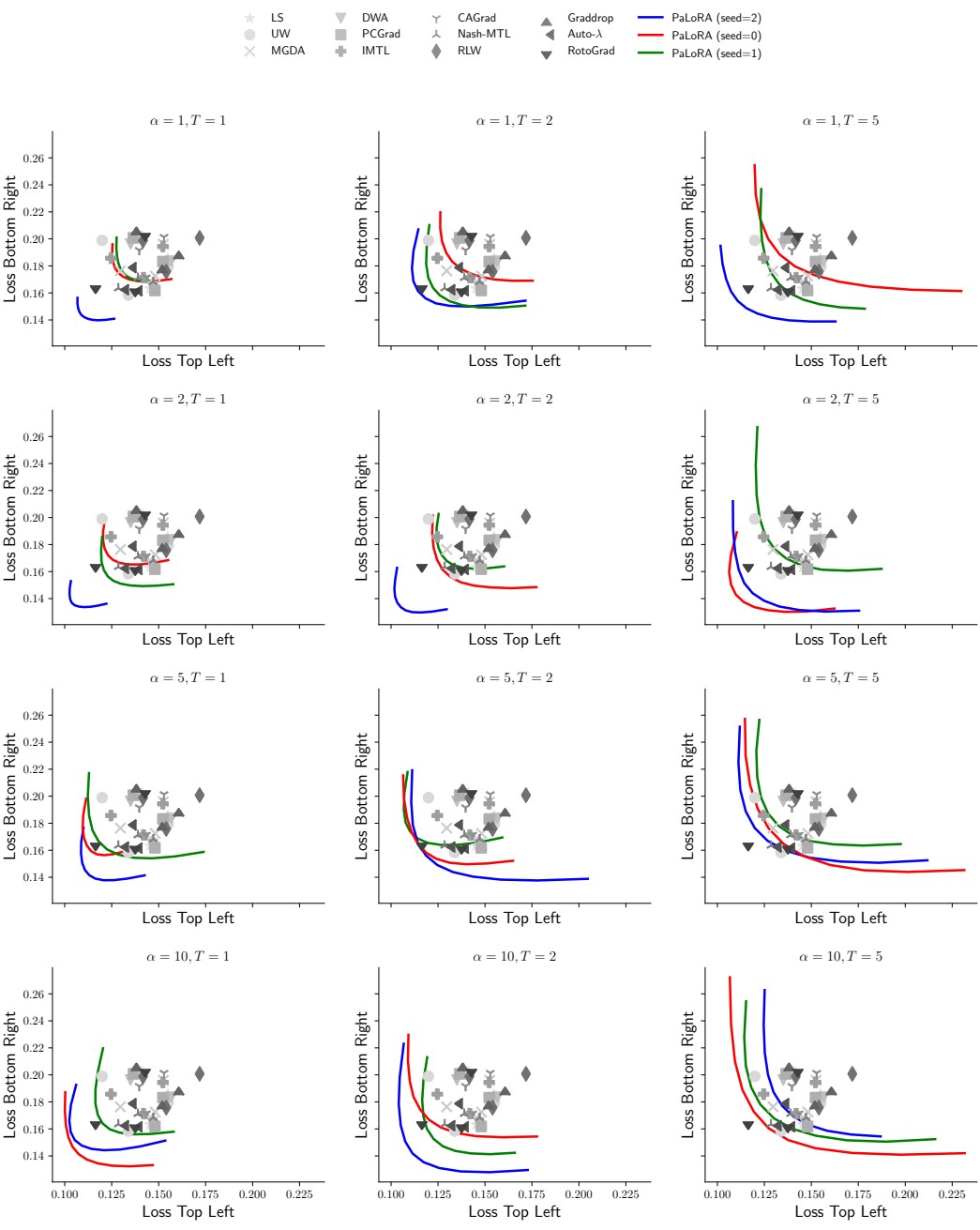

Figure 10: Validation losses for the setting outlined in Figure 9.

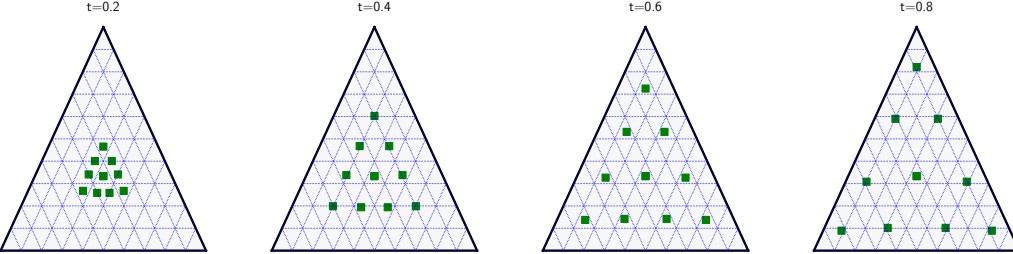

Figure 11: Examples of annealing in the case of 3 tasks for $m = 10$ points. Similar to the 2d case, the points are initially set in the center of the simplex and gradually towards the edges.

## C  DETERMINISTIC SCHEDULE

Section 4.2 discusses the deterministic sampling mechanism. In Equation 3, the preference vectors for a given timestep $\tau \in [0, 1]$ are a function of the base preferences $\{\boldsymbol{\lambda}_0^{(m)}\}_{m=1}^M$. For two tasks, we use torch.linspace(0, 1, M) to produce $\lambda$ in $[0, 1]$. The vector preference is then $\boldsymbol{\lambda} = [\lambda, 1-\lambda]$. For 3 tasks, we use the meshzoo library to produce the initial evenly distributed set in the simplex. Example for the 2d case are provided in Figure 2, while Figure 11 shows the case for three tasks.

## D  DETAILS OF THE ABLATION STUDY ON PREFERENCE SAMPLING

This section describes in greater detail the settings of the ablation study of Section 6.1. We ablate on the following dimensions:

1. number of forward pass $M \in \{3, 5, 7, 9\}$,
2. scaling $\alpha \in \{1, 2, 5, 10\}$, defined in Equation 2,
3. the following sampling schedules:
   (a) Deterministic sampling with annealing, defined in Equation 3. We explore temperature parameters $Q \in \{1, 2, 5\}$,
   (b) Deterministic sampling without annealing, defined in Equation 3 if $\tau = 1$ regardless of training iteration. We explore temperature parameters $Q \in \{1, 2, 5\}$,
   (c) Random sampling with annealing using the Dirichlet distribution $\text{Dirichlet}(p(1-\tau)\mathbf{1})$ for $p \in \{1, 2, 5\}$,
   (d) Random sampling without annealing using the Dirichlet distribution $\text{Dirichlet}(p\mathbf{1})$ for $p \in \{1, 2, 5\}$.

Prior works used solely schedule (a). We use 3 seeds per configuration and present the results as a scatter plot in Figure 12. Figure 12 shows the results focusing on the sampling mechanism only. Figure 13 and Figure 14 are supplementary to Section 6.1 of the main text, showing the effect of other hyperparameters as well.

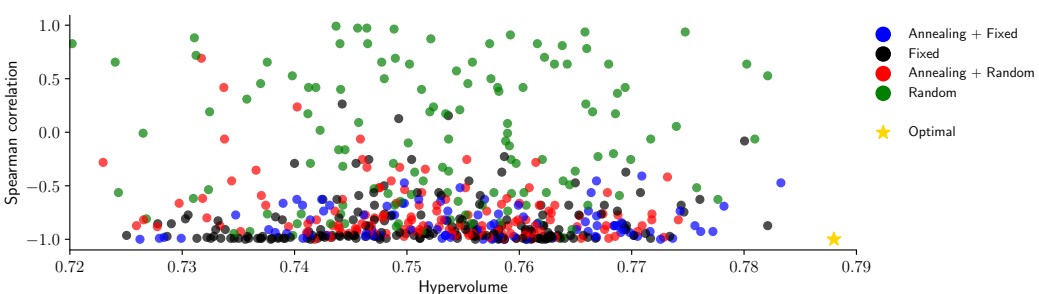

Figure 12: Ablation Study on the choice of preference vector sampling. Each scatter point corresponds to a different combination of scaling $\alpha$ and number of forward passes $M \in \mathbb{N}$. Deterministic schedules result in lower Spearman correlation reflecting more functionally diverse Pareto Fronts.

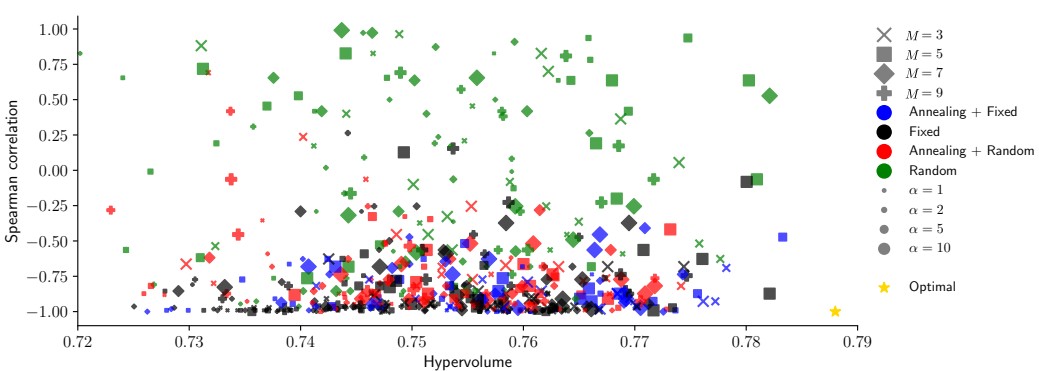

Figure 13: [Complement to Figure 12]: Ablation Study on the choice of preference vector sampling (in color) and the effect of scaling $\alpha$ (marker size) and number of forward passes $M \in \mathbb{N}$ (marker shape). Deterministic schedules result in lower Spearman correlation reflecting more functionally diverse Pareto Fronts.

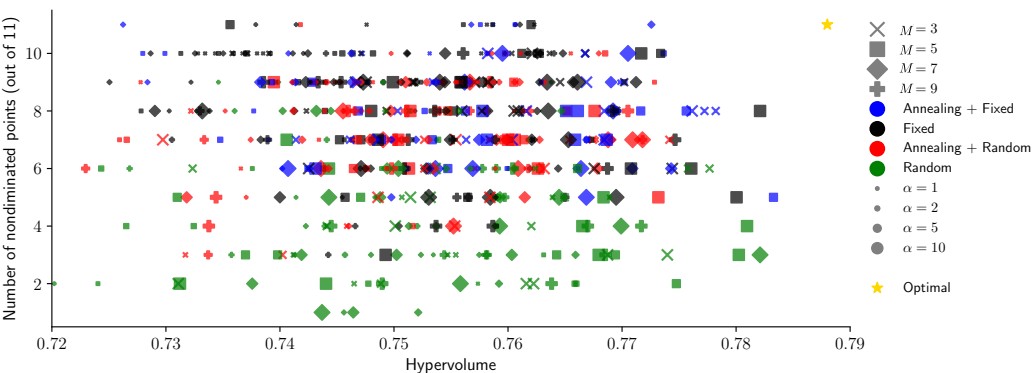

Figure 14: [Complement to Figure 13]: The y-axis changes to the number of nondiminated points.

## E    PROOF OF THEOREM 2

**Theorem 3.** *Let $f_t : \mathcal{X} \times \Theta \mapsto \mathcal{Y}$ be a family of continuous mappings, where $t = 1, \ldots, T$, and $\mathcal{X} \subset \mathbb{R}^D$ is compact. Then, $\forall \epsilon > 0$, there exists a ReLU multi-layer perceptron $f$ with three different weight parameterizations $\boldsymbol{\theta}_0, \boldsymbol{\theta}_1, \boldsymbol{\theta}_2 \in \Theta$, such that $\forall t \in [T]$, $\exists \alpha \in [0, 1]$, $\forall \boldsymbol{x} \in \mathcal{X}$:*

$$|f_t(\boldsymbol{x}) - f(\boldsymbol{x}; \boldsymbol{\theta}_0 + \alpha\boldsymbol{\theta}_1 + (1 - \alpha)\boldsymbol{\theta}_2)| \leq \epsilon.$$

The following proof is based on the proof provided by Dimitriadis et al. (2023).

*Proof.* Following the universal representation theorem, there exists $Q \in \mathbb{N}$, $\boldsymbol{M} \in \mathbb{R}^{(D+1)\times Q}$, $\boldsymbol{B} \in \mathbb{R}^Q$, $\boldsymbol{M'} \in \mathbb{R}^{Q \times D'}$ such that for a single hidden layer perceptron with non-linearity $\sigma$ such that:

$$g : A \times [0, 1] \to \mathbb{R}^{D'}$$
$$\boldsymbol{z} \mapsto \boldsymbol{M'}\sigma(\boldsymbol{M}\boldsymbol{z} + \boldsymbol{B}),$$

$$\forall \boldsymbol{x} \in A, \forall n \in \{1, \ldots, N\}, \left| f_n(\boldsymbol{x}) - g\left(x_1, \ldots, x_D, \frac{n-1}{N-1}\right)\right| \leq \epsilon.$$

For matrices

$$[\boldsymbol{R}]_{ij} = \begin{cases} 1 & i = 2j - 1 \\ -1 & i = 2j \\ 0 & \text{otherwise} \end{cases}$$

$$[\boldsymbol{S}]_{ij} = \begin{cases} 1 & j = 2i - 1 \\ -1 & j = 2i \\ 0 & \text{otherwise} \end{cases}$$

$$[\boldsymbol{U}_k]_i = \begin{cases} 0 & i \leq 2D \\ k & i = 2D + 1 \end{cases}$$

For $\boldsymbol{x} \in \mathbb{R}^D$, we have

$$\forall \alpha \geq 0, \quad \boldsymbol{S}\sigma(\boldsymbol{R}\boldsymbol{x} + \boldsymbol{U_0} + \alpha\boldsymbol{U_1} + (1-\alpha)\boldsymbol{U_2}) = (x_1, \ldots, x_D, \alpha).$$

For $\boldsymbol{\theta} = (\boldsymbol{R}, \boldsymbol{U}, \boldsymbol{M}\boldsymbol{S}, \boldsymbol{B}, \boldsymbol{M'})$, $\boldsymbol{\theta}_0 = (\boldsymbol{R}, \boldsymbol{U_0}, \boldsymbol{M}\boldsymbol{S}, \boldsymbol{B}, \boldsymbol{M'})$ and $\boldsymbol{\theta}_1 = (\boldsymbol{R}, \boldsymbol{U_1}, \boldsymbol{M}\boldsymbol{S}, \boldsymbol{B}, \boldsymbol{M'})$

$$f(\boldsymbol{x}; \boldsymbol{r}, \boldsymbol{u}, \boldsymbol{m}, \boldsymbol{b}, \boldsymbol{m'}) = \boldsymbol{m'}\sigma(\boldsymbol{m}\sigma(\boldsymbol{r}\boldsymbol{x} + \boldsymbol{u}) + \boldsymbol{b}),$$

then

$$\begin{aligned} f(x; \boldsymbol{\theta}_0 + \alpha\boldsymbol{\theta}_1 + (1-\alpha)\boldsymbol{\theta}_2) &= f(\boldsymbol{x}; \boldsymbol{R}, \boldsymbol{U_0} + \alpha\boldsymbol{U_1} + (1-\alpha)\boldsymbol{U_2}, \boldsymbol{M}\boldsymbol{S}, \boldsymbol{B}, \boldsymbol{M'}) \\ &= \boldsymbol{M'}\sigma(\boldsymbol{M}\boldsymbol{S}\sigma(\boldsymbol{R}\boldsymbol{x} + \boldsymbol{U_0} + \alpha\boldsymbol{U_1} + (1-\alpha)\boldsymbol{U_2}) + \boldsymbol{B}) \\ &= g(\boldsymbol{S}\sigma(\boldsymbol{R}\boldsymbol{x} + \boldsymbol{U_0} + \alpha\boldsymbol{U_1} + (1-\alpha)\boldsymbol{U_2})) \\ &= g(x_1, \ldots, x_D, \alpha) \end{aligned}$$

$\square$

