# OpenReview forum: "Pareto Low-Rank Adapters: Efficient Multi-Task Learning with Preferences"
_ICLR.cc/2025/Conference — ICLR 2025 Poster_

### Official Review · Reviewer_EQhb · 2024-11-01

**Soundness:** 2
**Presentation:** 3
**Contribution:** 2
**Rating:** 6
**Confidence:** 4

**Summary:**

PaLoRA presents a method for Pareto Front Learning (PFL) with two main contributions. First, the paper adds a low-rank adapter for each task, significantly reducing the overhead of the additional parameters added to the network. Second, they perform a deterministic (either fixed or annealing) preference sampling, leading to more stable learning and faster convergence.

**Strengths:**

- The proposed method is more efficient regarding number of parameters, i.e., memory complexity, compared to PaMaL and outperforms it as well and they also scale up to larger number of tasks (as shown with the SARCOS).
- Interesting discussion sections that shows the efficiency of their method in converging faster and also the multiple adapters learning different tasks.

**Weaknesses:**

- What happens if you train longer for the SARCOS dataset? I.e., Figure 3 (c) goes for more than 100 epochs (until convergence of each model)? How does PaLoRA compare to PaMaL?
- Have you done a ablation over different ranks for the LoRA modules? It would be great if you could report the numbers for different ranks.
- Please provide the standard deviation for the runs in Table 1 and also for the numbers for MultiMNIST (given the variance among the performances of PaLoRA on MultiMNIST and MultiMNIST-3).
- Would be great to have another dataset added to the experiments, such as  UCI Census-Income.
- Figure 3 (b) could improve and have the legends in the image.

**Questions:**

Please refer to the "weaknesses".

---

> ### Author Response · Authors · 2024-11-22
>
> We thank reviewer EQhb for their constructive feedback. We are glad that they find our discussion interesting and the algorithm more memory efficient and faster than baselines. Additional experiments have been added to the appendix and changes in the manuscript are in blue font. We address their points below.
>
> **Training Sarcos for longer**
>
> Thank you for the suggestion. We have repeated the same experiment for Figure 3(c) but now train for 500 epochs. The results are presented in Figure 14 of the new Appendix F. Again, we see that PaLoRA outperforms PaMaL and converges faster, even for lower number of forward passes $M$.
>
>
> **Ablation on the ranks**
>
> We have added an ablation over the different ranks in the appendix, in Table 3.. We have used the Cityscapes benchmark and considered ranks $r\in\{1,2,4,8\}$, our proposed fixed deterministic schedule and $M\in\{3,5\}$. We use 3 seeds per combination. Rank 1 results in an excess of approximately 1.05% of the parameters of the MTL model. The memory requirements for the other cases scale linearly with the rank. We see that the method is robust to the choice of rank.
>
>
> **Providing standard deviations**
>
> MultiMNIST-3 is more difficult compared to MultiMNIST, as MultiMNIST is more difficult compared to MNIST. The reason is that in the construction of MultiMNIST, there is an overlap between the digits. This overlap is further increased in the case of MultiMNIST-3, resulting in lower accuracies. We have provided the standard deviations for MultiMNIST in the Appendix. The baseline numbers for Table 1 are drawn directly from [1] and, therefore, we do not have the standard deviations. However, we have added the standard deviation for NYUv2 (Table 2) in the new appendix.
>
>
> **Experiment on UCI-Census**
>
> We did not consider UCI-Census since the network used in [1] is a single-layer MLP with 256 neurons. Therefore, parameter-efficient approaches such as PaLoRA have no added value to such small settings. To counterbalance the lack of Census, we have included the rest of the experiments from [1], namely MultiMNIST, MultiMNIST-3, UTKFace and Cityscapes following their implementation. Furthermore, we have expanded beyond the experimental settings of [1] to include the most challenging benchmark in NYUv2 as well as SARCOS which has 7 tasks. In both cases, we show that PaLoRA outperforms the current PFL state-of-the-art PaMaL.
>
> **Weakness 5**
> Thank you for drawing our attention to this. We have updated Figure 3b.
>
> We hope that our response has addressed the reviewer’s comments. We kindly ask them to reconsider their score.
>
> [1] Dimitriadis N, Frossard P, Fleuret F. Pareto Manifold Learning: Tackling multiple tasks via ensembles of single-task models. ICML 2023

---

> > ### Author Response · Authors · 2024-12-01
> >
> > Dear Reviewer EQhb,
> >
> > As the rebuttal phase is nearing its conclusion, we would like to ask if our responses have addressed your concerns. If so, we would greatly appreciate it if you could reconsider your score.
> >
> > Kind regards,
> >
> > the authors

---

> > > ### Comment · Reviewer_EQhb · 2024-12-02
> > > **Resposne**
> > >
> > > Dear authors,
> > >
> > > Thank you for considering the comments and answering them all and providing the additional experiments and clarifications.
> > >
> > > Regarding the ablation on the number of ranks, it seems *all* of the settings are better than the baseline models (even rank 1 with M=3) and even the PaLoRA model reported in Table 1. Is there a reason that:
> > >  - **you are not using the rank=1 M=3 setting in Table 1**, given that it requires less parameters?
> > >  - the same PaLoRA setting (rank=4 M=5) has a huge (51.27 vs. 41.22) **difference in the "Depth Rel Err." between Table 1 and 3 on the same dataset**? Does this make sense?
> > >
> > >
> > > Given the changes made by the authors and the plots added, I would like to update my score.

---

> > > > ### Author Response · Authors · 2024-12-02
> > > >
> > > > Dear reviewer,
> > > >
> > > > We sincerely appreciate the increase in your score, and we thank you.
> > > >
> > > > Regarding the ablation on the rank, the differences are mainly due to the different seeds used between the two sets of results.  Apart from this change, “Depth Rel Err” exhibits more variability compared to “Depth Abs Err”, as noted in prior works as well. These two factors explain the observed difference. Concerning the results in Table 1, we selected r=4 after some initial ablations. Following the reviewer’s suggestions and our novel findings indicating that lower ranks can result in competitive performance with even lower memory cost, we are repeating the experiment in Table 1 using lower ranks and will add the results to the paper.
> > > >
> > > > Kind regards,
> > > >
> > > > the authors

---

### Official Review · Reviewer_amQG · 2024-11-03

**Soundness:** 3
**Presentation:** 3
**Contribution:** 2
**Rating:** 5
**Confidence:** 3

**Summary:**

This paper proposes Pareto Low-Rank Adaptors (PaLoRA), which are claimed to address the problem of limited scalability, slow convergence, excessive memory requirements, and inconsistent mappings from preference to objective space that existed in previous methods. The goal is achieved by adding multiple task-specific low-rank adapters and optimizing the weights with Pareto Front. Experiments on multiple tasks and variant datasets show the effectiveness of the proposed method.

**Strengths:**

1. This paper proposes to address multi-task learning with multiple low-rank adapters equipped with Pareto Front Learning.

**Weaknesses:**

1. It lacks discussions between PaLoRA and MoE-like methods.
2. It lacks discussions between PaLoRA and general adaptive weight learning methods. How is PaLoRA better and why?
3. It seems that this work applies PFL on multiple Lora, any significant differences between general PFL applications?
4. Comparisons between GFLOPS/Memory/Speed with other methods at the inference stage?
5. What is "Functional Diversity" in Sec 4.2? Any definitions or quantitative evaluations?
6. Any comparisons between PaLoRA and other LoRA-using methods on multi-task learning?
7. It points out the problem of " inconsistent mappings from preference to objective space", but no experiments and explanations validate the improvements or the relationship between mapping consistency and good performance/generalization on multiple tasks.

**Questions:**

See weaknesses.

---

> ### Author Response · Authors · 2024-11-22
>
> We thank the reviewer for their feedback. Additional experiments have been added to the appendix and changes in the manuscript are in blue font. We reply to the points raised below.
>
> **Discussion between PaLoRA and MoE-like methods**
>
> We appreciate the reviewer’s suggestion to discuss comparisons with MoE-like methods. First, MoEs are typically employed in very large-scale models (e.g., foundation models) to handle heterogeneous inputs or datasets, often relying on sparse routing mechanisms. In contrast, PaLoRA is designed for end-to-end training with multi-task settings where all tasks originate from the same input but the supervision is different (e.g. in Cityscapes the scene of a road is the input and the tasks are semantic segmentation and depth regression). Therefore, PaLoRA is input agnostic while MoEs are input-specific. Moreover, unlike MoEs, which make discrete routing choices (topk routing), our method explicitly aims for a continuous parameterization of the Pareto Front to support fine-grained control and flexibility across objectives. Finally, by integrating LoRAs into the core model during inference, PaLoRA maintains the same memory and inference costs as the base model, whereas MoEs inherently introduce additional computational overhead and memory due to the routing mechanism. That’s the reason why we do not discuss MoEs in the paper.
>
>
> **Discussion between PaLoRA and adaptive weight learning methods**
>
> We have already compared against several baselines that can be considered adaptive weight learning methods. For instance, 9 out of 12 MTL baselines (all excluding LS, RLW and Graddrop) could be considered as adaptive. However, they are different from our approach since they only find one point in the Pareto Front, while we (and Pareto Front Learning methods in general) continuously parameterize the entire Pareto Front in one training run. Therefore, compared to the MTL baselines, PaLoRA has a more general and difficult goal. Nonetheless, our experiments show our method performs at par or better, while remaining efficient. Finally, our Pareto Front Learning baselines can also be considered adaptive weight learning methods and differ in their weight generating mechanism, e.g., [1] uses hypernetworks, [2] uses weight ensembles, while we used task-specific LoRAs. Beyond the much lower memory requirements of our method due to this architectural change, we are also the first to address the sampling mechanism underlying Pareto Front Learning and we show it plays a crucial role, leading to superior results.
>
>
> **Differences with general PFL applications**
>
> Can the reviewer please clarify what they mean by general Pareto Front Learning (PFL) *applications*? Past works on Pareto Front Learning such as [1] and [2] differ in their parameterization of the Pareto Front; [1] uses a hypernetwork while [2] a weight ensemble. Our choice to train LoRAs and one copy of the original model (instead of multiple model copies like [2]) significantly reduces the number of parameters and helps in optimization, e.g., for NYUv2 benchmark in Table 2 our parameter excess if 6.2% and for [2] it is 200% while we achieve better results.
>
> Besides the architectural component, we are the first to focus on the sampling component of Pareto Front Learning (§4.2) where we show that it really improves both convergence and validity of the final Pareto Front (in the entire experimental section and in more detail in §6.1). Overall, these contributions achieve state-of-the-art results, while being more efficient.
>
>
>
> **GFLOPS/memory comparison**
>
> During inference, if a preference ray has been selected by the user, PaLoRA and all other methods we consider have exactly the same GFLOPS/memory requirements. If a preference ray has not been selected, the target model must first be constructed for PFL methods, resulting in additional memory and computation. However, the construction of a model does not need to be performed each time, but only when the user wishes to change the trade-off of the model.
>
> For the comparison, we include an experiment on SegNet with Cityscapes, assuming a batch size of 10. We compare only with PaMaL as a PFL method, since PHN and COSMOS do not scale to this benchmark.
>
> In terms of memory:
> - MTL Model size in GPU: 100.13 MB (this is the same for PaLoRA and PaMaL if the preference is set)
> - PaLoRA model size in GPU: 104.77 MB (rank=4 as in Table 1)
> - PaMaL model size in GPU: 200.26 MB (duplicate memory because we have two tasks)
>
> The forward pass for all MTL models is:
> - MTL Forward pass: 15.0716 ms
>
> While the construction times for the PFL methods that scale to this benchmark are:
> - PaLoRA model construction: 3.9677 ms
> - PaMaL model construction: 1.6577 ms

---

> > ### Author Response · Authors · 2024-11-22
> >
> > **Clarifying Functional diversity**
> >
> > Functional diversity refers to the quality of the solution set produced by a PFL method in terms of Pareto properties, as defined in Definition 1. Prior works have used the Uniformity metric [1] or Spearman correlation [2] to capture the monotonicity of the final solution set. In our case, we simplify the metric and simply present how many points in the solution set actually lie in the Pareto Front. For the case of two tasks, all PFL methods are evaluated by sampling 11 points corresponding to preferences $[\lambda, 1-\lambda], \lambda \in \{0, 0.1, \dots, 0.9, 1\}$. These 11 evaluations result in the solution set of each method. Figures 3a, 4, 5b, 11, 12, 13 show that PaLoRA consistently produces “functionally diverse” Pareto Front, while other methods suffer. In particular, Figure 5b compares with SOTA PaMaL [2] and shows that PaLoRA maintains an almost perfect Pareto Front (11 out of 11 points) throughout training while PaMaL or PaLoRA without the deterministic sampling component fluctuate a lot. Figures 11, 12 and 13 further highlight the superiority of PaMaL in a large ablation study and include Spearman correlation as a metric. Overall, our method is consistent in producing a valid Pareto Front. We hope this clarifies the reviewer’s concern.
> >
> >
> >
> >
> > **Comparison with LoRA-MTL method**
> >
> > Given your and reviewer vwtQ’s comments, we have added [3,4,5,6] in our related work section, discussing the difference between our work and other works using LoRA in a multi-task setting. Compared to [3,4,5,6], PaLoRA trains from scratch (we do not fine-tune from a pre-trained model with the exception of Section 5.4) and not towards a single point in objective space but towards continuously parameterizing the Pareto Front. Therefore, existing LoRA-based methods cannot be compared directly with PaLoRA due to different settings. For this reason, we have followed the experimental protocol of the current PFL state-of-the-art [2] and considered 3 PFL baselines and 12 MTL baselines.
> >
> > **Clarifying “inconsistent mappings from preference to objective space”**
> >
> > The “inconsistent mappings from preference to objective space” apply only to PFL methods and have been explained in §4.2 . We showed this in the experimental section, such as Figure 3a and Figure 5b. Given a user’s preference, PFL methods generate the weights of a neural network that upon evaluation maps to a point in objective space. By repeating this process over a continuous domain of preferences, e.g., for two tasks going over $[\lambda, 1-\lambda], \lambda\in[0,1]$, each PFL method constructs a Pareto Front. Our goal is for this mapping to be consistent (monotonic), i.e., putting more emphasis on task 1 (higher $\lambda$) results in an improvement for task 1 at the expense of task 2. For example, Figure 3 shows that other PFL methods (presented as lineplots) result in inconsistent mappings (for PHN [1] and COSMOS) and decreased performance compared to us for PaMaL [2]. Finally, Figure 5b shows that mappings are consistent throughout training and the superiority of the mappings produced by PaLoRA is further explored in Figures 11,12,13. Therefore, we believe that out statement is appropriately backed up by the experiments.
> >
> > We hope that our response has addressed the reviewer’s concerns and, if yes, we kindly ask the reviewer to reconsider their score.
> >
> > [1] Navon A, Shamsian A, Chechik G, Fetaya E. Learning the pareto front with hypernetworks. ICLR 2021
> >
> > [2] Dimitriadis N, Frossard P, Fleuret F. Pareto Manifold Learning: Tackling multiple tasks via ensembles of single-task models. ICML 2023
> >
> > [3] Mixture-of-Subspaces in Low-Rank Adaptation, EMNLP 2024.
> >
> > [4] Loraretriever: Input-aware lora retrieval and composition for mixed tasks in the wild, ACL 2024.
> >
> > [5] HydraLoRA: An Asymmetric LoRA Architecture for Efficient Fine-Tuning, NeurIPS 2024.
> >
> > [6] Mixture-of-LoRAs: An Efficient Multitask Tuning for Large Language Models, COLING 2024.

---

> > > ### Author Response · Authors · 2024-12-01
> > >
> > > Dear Reviewer amQG,
> > >
> > > As the rebuttal phase is nearing its conclusion, we would like to ask if our responses have addressed your concerns. If so, we would greatly appreciate it if you could reconsider your score.
> > >
> > > Kind regards,
> > >
> > > the authors

---

### Official Review · Reviewer_9e2E · 2024-11-04

**Soundness:** 3
**Presentation:** 3
**Contribution:** 2
**Rating:** 6
**Confidence:** 4

**Summary:**

The paper presented a multi-task learning method using low-rank adaptors. The paper formulates the multi-task learning problem as Pareto Front Learning, where points on the Pareto Front are sampled and optimised in training. Different from previous work which uses random sampling for such point, the paper proposed to use fixed interval, sampled either evenly or using a pre-defined schedule. Across different benchmarks, the method outperforms other Pareto Front Learning methods, while sometimes underperforms conventional multi-task learning methods.

**Strengths:**

1. The proposed method uses low-rank adaptors to encode weight updates in multi-task learning, and is more efficient compared to previous work. Specifically, it achieves over $\times 20$ reduction of memory usage. This is particularly impactful in real-world applications, where large neural networks are typically memory-intensive and costly to scale.

2. The proposed deterministic sampling of preference, i.e., $\lambda$, seems like a notable improvement over the existing random sampling strategy. The annealing strategy is fairly well motivated. By gradually moving $\lambda$ from the centre of the simplex to the faces, the sampled point transition from general models to more task-specific ones. This was shown to increase the convergence speed and stability.

3. The paper presented fairly comprehensive experiments across multiple benchmarks and demonstrates the superior performance of the proposed method over existing PFL methods, despite the occasional lower performance compared to MTL methods. The paper also includes rich visualisations that very well illustrates the ideas behind the method and the experimental results.

**Weaknesses:**

1. Being able to construct a Pareto Front of models is in itself an interesting result. But what is the practical value of this? The objective of multi-task learning is to produce one model that performs well on two or more tasks. By this definition, it seems sufficient to just have one model, or rather one point on the Pareto Front. I think it will make the paper much stronger if some practicality can be demonstrated, besides the nice theoretical result. Perhaps the Pareto Front can help in OOD scenarios. For instance, after a model is trained on cityscapes for segmentation and depth estimation, when it is tested on MS COCO for segmentation, I imagine MTL methods might be less effective due to the distribution shift, but the proposed method can still adjust the coefficients/preference to the new dataset.

2. The contribution of the fixed schedule is not the most convincing. The disadvantage of random sampling according to the paper seems to be that it produces a tight cluster of sampled points. This may be related to the distribution used rather than the random sampling process. In addition, there doesn't seem to be any experiments showing the improvement of the annealed schedule compared against the even-sampling strategy.

3. The motivation of the annealed strategy seems to be mimicking the training dynamics, where the network learn generic features in early stages while specialise in later stages. I don't think this is a homogeneous behaviour across the entire model. Usually the shallow layers of model extract generic, low-level features while the deeper layers extract task-specific features. Perhaps it makes more sense to apply this annealed schedule layer wise, instead of chronologically. There is in fact an increasing body of work on multi-task learning/model merging using learned layer-wise coefficients, such as Zhang et al. (2024) and Yang et al. (2024). I think the annealed schedule will make more sense in this context.

Refs:

- Zhang et al., Knowledge Composition using Task Vectors with Learned Anisotropic Scaling. NeurIPS 2024
- Yang et al., AdaMerging: Adaptive Model Merging for Multi-Task Learning. ICLR 2024

**Questions:**

1. Why does PaLoRA has a memory overhead compared against the MTL baseline? If I understand this correctly, only the LoRAs are trained, albeit one for each task. But training a LoRA only costs a fraction of the memory required for fine-tuning the whole model, which is needed for MTL if I didn't misunderstand the settings.

2. For M>1, is it essentially doing gradient accumulation? Otherwise what does multiple forward passes refer to?

---

> ### Author Response · Authors · 2024-11-22
>
> We thank reviewer 9e2E for their constructive feedback. We are glad that they find our deterministic sampling a notable improvement, our use of low-rank adaptors relevant to real-world applications, and our visualizations rich.
> Additional experiments have been added to the appendix and changes in the manuscript are in blue font. We address their points below.
>
> **On the practical benefits of Pareto Front Learning**
>
> Thank you for the suggestion. We have added some application domains in the second paragraph of the introduction to highlight the practical benefits of Pareto Front Learning.  Indeed, Multi-task learning seeks to produce a single model. However, constructing the Pareto Front (PF) provides flexibility for scenarios where task priorities shift dynamically, or the user-trade-offs are not known in advance. For example, traffic conditions in autonomous driving may demand trade-offs between segmentation and depth estimation. This flexibility is valuable across diverse applications. The PFL practical benefits can also be traced back to the academic interest in the topic with Multi-objective optimization algorithms such as NSGA-II [6] (with over 50k citations) addressing smaller scale settings using genetic algorithms and Pareto Hypernetworks [3] adapting the construction of a continuous Pareto Front in neural networks. Finally, a recent work [6] (to appear in NeurIPS) focuses on adapting a single LLM to a broad spectrum of human preferences in a Pareto-optimal manner, while [7] focuses on multi-objective generation.
>
> Our experimental setup, therefore, is based on the PFL literature and actually expands on it to show the scalability of our method. For instance, [3] performs experiments up to Cityscapes but with small networks (ENet), [4] scales up to networks like SegNet and we further explore NYUv2, a more difficult benchmark with 3 tasks instead of 2. For these reasons, OOD scenarios are beyond the scope of the paper. We hope that our response and the addition of application domains in the introduction of the manuscript have clarified the benefits of PFL.
>
>
>
> **Random vs deterministic schedule and Fixed vs Annealed**
>
> Modifying the random sampling distribution inevitably introduces hyper-parameters, while a fixed schedule does not. Furthermore, even if a better distribution is selected, the stochasticity can again lead to pernicious updates, discussed in Section 4.2, due to an imbalanced concentration of sampled rays. This is particularly important for lower values of $M$ (the number of forward passes). Our ablation study in the appendix (Figures 11, 12 and 13) show that a fixed deterministic schedule results in a more valid and functionally diverse Pareto Front (y axis for all plots) compared to random sampling. Regarding the benefits of annealed vs fixed deterministic sampling: annealed sampling allows for lower values of M (marker shape in the aforementioned plots) which translates to lower computational requirements. Finally, it is important to note that the major performance boost comes from replacing the random sampling with a deterministic one, while an annealed deterministic schedule compounds on these benefits.
>
>
>
> **On the possibility of layer-wise scaling**
>
> Indeed the behavior is not homogenous across the entire model, as a long line of literature shows that low-level features lie in early layers and task-specific on deeper layers. In our setting, it is imperative that each task is associated with a *scalar* coefficient $\lambda_t$ because we use the same scalar to weigh the loss for the same task $\mathcal{L}_t$. Doing so steers the model towards the desired Pareto properties and, during inference, users can specify their desired trade-off and get the associated model weights. In contrast, works like [1,2] focus on attaining a single model in a post-training setting.
>
> It would be really interesting to check whether in this after-training setting we can merge two models towards constructing a Pareto Front simply by looking at the coefficients. In that setting, intuitively, an annealed schedule on the layers does make sense.
>
> Regarding layer-wise scaling during *training*, preference rays are sampled from T-dimensional simplex
> $\Delta_T=\{{\lambda}\in\mathbb{R}^T_+: \sum_{t=1}^T\lambda_t=k\}$  for $k=1$. Per the reviewer’s suggestion, applying layer-wise scaling  during *training* could be done by adding a schedule on $k$ dependent on the layer. The simplest could be linear, i.e., for layer $\ell$ out of $L$, the “modified” simplex will use $k_\ell=\frac{\ell}{L}$. Effectively, lower layers will use a lower range in the task-specific ray coefficients. However, the network weights may naturally adapt making the layer-wise schedule not needed at all or more complex schedules might be needed for the layer-wise scaling to work. This is an interesting idea, but introduces multiple hyperparameters, and therefore goes beyond the scope of our work.

---

> > ### Author Response · Authors · 2024-11-22
> >
> > **Question 1**
> > PaLoRA focuses on training end-to-end all the parameters (original network and LoRAs). This is the architectural contribution of our work that departs from using hypernetworks [3] or weight ensembles [4]. Therefore, it reduces by orders of magnitude the additional parameters while achieving better performance. Since LoRAs were originally introduced as a fine-tuning method, we do include experiments in §5.4 where we only fine-tune the LoRAs and treat the backbone as a pre-trained model. All other experiments follow the Pareto Front Learning literature, e.g., [3,4], and train end-to-end.
> >
> >
> > **Question 2**
> > Despite both requiring multiple forward passes, M>1 and gradient accumulation are different. Gradient accumulation uses the same model, and each batch is different. Instead M>1 uses a different model (defined by different merging coefficients in Equation 2) but with the same data. Since the model changes a lot at each batch based on the sampled preference ray $\lambda$, performing multiple passes results in higher stability, especially if the proposed deterministic schedule is used. Full details for the case M>1 are provided in §4.2.
> >
> > [1] Zhang et al., Knowledge Composition using Task Vectors with Learned Anisotropic Scaling. NeurIPS 2024
> >
> > [2] Yang et al., AdaMerging: Adaptive Model Merging for Multi-Task Learning. ICLR 2024
> >
> > [3] Navon A, Shamsian A, Chechik G, Fetaya E. Learning the pareto front with hypernetworks. ICLR 2021
> >
> > [4] Dimitriadis N, Frossard P, Fleuret F. Pareto Manifold Learning: Tackling multiple tasks via ensembles of single-task models. ICML 2023
> >
> > [5] Deb K, Pratap A, Agarwal S, Meyarivan TA. A fast and elitist multiobjective genetic algorithm: NSGA-II. IEEE transactions on evolutionary computation. 2002 Apr;6(2):182-97.
> >
> > [6] Zhong Y, Ma C, Zhang X, Yang Z, Chen H, Zhang Q, Qi S, Yang Y. Panacea: Pareto alignment via preference adaptation for llms. arXiv preprint arXiv:2402.02030. 2024 Feb 3.
> >
> > [7] Yao Y, Pan Y, Li J, Tsang I, Yao X. PROUD: PaRetO-gUided diffusion model for multi-objective generation. Machine Learning. 2024 Sep;113(9):6511-38.

---

> > > ### Comment · Reviewer_9e2E · 2024-11-28
> > >
> > > Thank you for the detailed response. The rebuttal has addressed most of my questions. So I'm still leaning towards accepting the paper.
> > >
> > > Nevertheless, I think the technical contribution and the practicality of the proposed method are not as strong as it could be. The main technical contribution is the annealed deterministic schedule. But as the authors also pointed out, the performance improvement primarily comes from simply replacing random sampling with deterministic sampling. The impact of the annealing is less pronounced. In addition, without directly demonstrating the practicality of the Pareto Front, it is hard to gauge the impact the work in real-life applications. Although I do appreciate the authors providing more context of PFL and how it may be helpful in applications like autonomous driving, I think more tangible results need to be presented for it to be more convincing.
> > >
> > > In conclusion, while I believe the paper is above the acceptance threshold, I do not think it is strong enough to merit an increase in rating.

---

> > > > ### Author Response · Authors · 2024-11-30
> > > >
> > > > Dear reviewer 9e2E,
> > > >
> > > > We are writing again to clarify the contribution regarding the deterministic sampling. We disagree with the reviewer that the main technical contribution is the *annealed* deterministic schedule. In fact, as outlined, in the abstract, introduction and Section 4 (which introduces the method), our contributions are:
> > > >
> > > > 1. Parameterization of the Pareto Front in the convex hull of LoRAs. We restate that, despite its popularization as a fine-tuning mechanism, we employ low-rank adapters when training from scratch and actually address a crucial limitation of the state-of-the-art, PaMaL [1], in terms of memory overhead; we need less memory, achieve better results and faster convergence
> > > > 2. Deterministic sampling: Our proposition is to use a deterministic sampling instead of the random that dominates the literature.**The annealing is only a secondary part of this contribution**. We believe that we have adequately shown that fixed sampling works much better than random. This is why the title of the subsection that introduces the deterministic sampling omits the word “annealed”.
> > > >
> > > > Furthermore, we should clearly not be the one “directly demonstrating the practicality of the Pareto Front”, as it is a completely standard and well established field. A weakness such as this invalidates the entire field and the efforts of many researchers.
> > > >
> > > > In light of these clarifications, we kindly ask the reviewer to reconsider their score.
> > > >
> > > > Kind regards,
> > > >
> > > > the authors
> > > >
> > > > [1] Pareto Manifold Learning: Tackling multiple tasks via ensembles of single-task models. ICML 2023

---

> > > > > ### Comment · Reviewer_9e2E · 2024-12-02
> > > > >
> > > > > Thanks for the further clarification.
> > > > >
> > > > > I understand and appreciate the contributions this paper has made. Perhaps the rating scale doesn't reflect this very well. In other occasions, I'd increase the rating to 7/10, which is unfortunately not available in the scale. I can see the work being valuable to a niche community. But without tangible results beyond it, it is hard to gauge its impact.

---

### Official Review · Reviewer_vwtQ · 2024-11-04

**Soundness:** 2
**Presentation:** 3
**Contribution:** 2
**Rating:** 5
**Confidence:** 3

**Summary:**

This paper introduces a new method called PaLoRA (Pareto Low-Rank Adapters) to address efficiency issues in multi-task learning. PaLoRA enhances any neural network architecture by adding task-specific low-rank adapters and continuously parameterizing the Pareto frontier within their convex hull. This approach encourages the original model and adapters to learn general features and task-specific features, respectively, thereby improving the model's flexibility and efficiency.
Additionally, the paper proposes a deterministic sampling schedule. This schedule reinforces the division of labor between tasks, accelerates convergence, and strengthens the validity of the mapping from preference vectors to the objective space. Specifically, the sampling schedule evolves over training time, gradually shifting the focus from learning general features to learning task-specific features, ensuring the effectiveness and stability of the training process.

**Strengths:**

1. This paper is well-organized and clearly written.
2. The idea of decouple a neural network into a general feature extrator and several task-specific low-rank adapters is reasonable.

**Weaknesses:**

1. The key equation (2) is quite similar to LoRA (Low-Rank Adaptation). And there are many multi-LoRA methods proposed for multi-task learning. However, the authors have missed. Please tell us the benefits of the proposed method compared with the multi-LoRA methods (e.g., [1-4]).
2. Fig.2 is difficult to understand. First, how is $\lambda$ changed over time? Second, how to judge the quality of the mappings from preference to objective space?
3. For Fig.3, why are there fewer or even no compared methods in (b) and (c)?
4. In Table 1 and Table 2, the proposed PaLoRA performs similarly to the other MTL methods. If the main purpose is accelerating convergence and reducing the memory requirements, then, I think it should be compared with the multi-LoRA methods [1-4].
5. The effects of the hyper-parameters (e.g., $\alpha$, $r$, and $Q$) on the results are not well studied in the experiments.

[1] Mixture-of-Subspaces in Low-Rank Adaptation, EMNLP 2024.
[2] Loraretriever: Input-aware lora retrieval and composition for mixed tasks in the wild, ACL 2024.
[3] HydraLoRA: An Asymmetric LoRA Architecture for Efficient Fine-Tuning, NeurIPS 2024.
[4] Mixture-of-LoRAs: An Efficient Multitask Tuning for Large Language Models, COLING 2024.

**Questions:**

Please see the weaknesses.

---

> ### Author Response · Authors · 2024-11-22
>
> We thank the reviewer for their feedback. We are glad that they found our paper well written and appreciated our idea. Additional experiments have been added to the appendix and changes in the manuscript are in blue font. We address their comments below.
>
> **Related works with LoRA**
>
> Thank you for drawing our attention to these works. We have included them in our updated related work section. It is important to note that [1,2,3] are considered contemporaneous works according to [ICLR guidelines](https://iclr.cc/Conferences/2025/ReviewerGuide), since [1] was presented in November (after submission), [2] in August and [3] will be presented in December. Our proposed method is materially different to all these approaches in both settings and objectives. Specifically, we train both LoRAs and the original network from scratch (with the exception of §5.4) while [1,2,3,4] focus on fine-tuning from foundation models.
>
> In terms of objective, we focus on parameterizing the entire Pareto Front with one network, but
> - [4] focuses on improving fine-tuning towards a single solution, similar to all our MTL baselines, by learning a routing mechanism for multiple already trained LoRAs,
> - [2] on dynamically composing already trained LoRAs and ,
> - [1,3] are direct LoRA replacements and not specific to multi-task scenarios.
>
> For these reasons, none of these works are comparable with our setting. We hope that our comment clarifies the vast differences with these approaches.
>
>
>
>
> **Clarification on Figure 2**
>
> We have updated the caption to make the plot easier to understand. This plot explains how the sampling works between previous works (subfigures a, b) and our proposed deterministic sampling (subfigures c, d). Each dashed line represents the sampling of preference rays for one batch and the arrow on the left shows the direction of time. Following [5], depicted in subfigure b, we sample multiple preference rays $\lambda$ per batch and perform as many forward passes, but each time the weights of the model change according to Equation 2. The plot shows the case for two tasks, where the preference ray is $[\lambda, 1-\lambda], \lambda\in[0,1]$. Therefore, we only depict the weight for the first task in Figure 2. The plot does not present any experimental results, i.e., we do not show the quality of the produced mappings in this plot. The quality of the mappings can be seen in Figures 3ab, 4 and (quantitatively) in Figures 5b, 11, 12, 13 where we show that the proposed *deterministic* sampling results in more valid Pareto Fronts. We hope that this has clarified the reviewer’s questions.
>
>
> **On the number of compared methods of Figure 3**
>
> Each subplot of Figure 3 presents a different perspective on the results. Figure 3b offers a qualitative result showing that PaLoRA can retrieve a wide Pareto Front for 3 tasks. We do the same for another dataset in Figure 7 of the appendix, following PaMaL [5]. We believe that adding more PFL methods  would make the plots too dense and hard to parse for the readers. This is why we opted to present results as this. Therefore, we have only presented all the baselines on subfigure a, since the dataset has two tasks.
>
> For Figure 3c, we selected to compare PaLoRA with the state-of-the-art PFL method PaMaL [5] and use a single MTL method as a frame of reference. We selected the simplest one in Linear Scalarization. We did not add more MTL or PFL methods since the goal of this plot is to show how better PaLoRA scales compared to PaMaL for more tasks. We have expanded on the SARCOS experiment in the appendix for the case of longer training (500 instead of 100 epochs), where we observe again that PaLoRA scales better to PaMaL.
>
>
> **Clarification on the purpose of the method**
>
> It is important to note that [1,2,3] are considered contemporaneous works according to [ICLR guidelines](https://iclr.cc/Conferences/2025/ReviewerGuide), since [1] was presented in November (after submission), [2] in August and [3] will be presented in December. The main purpose is accelerating convergence and reducing memory requirements **only when compared to PFL methods**, such as PHN and PaMaL. Therefore, our method is on a different class of methodologies compared to MTL approaches, which find only one point in the Pareto Front. We have added two sentences to highlight this: a sentence in line 53 as well as a sentence in the beginning of Section 4.2. We do include standard multi-task learning methodologies to show that, despite our objective being more general and harder, we still compete or even outperform them. Our experiments and baseline selection follow the sota in PFL [5] but we push the boundary with the even more challenging benchmark NYUv2. Compared to [1-4], both settings (fine-tuning vs training from scratch) and objectives (one point in the Pareto Front vs continuous parameterization) are different, as explained in more detail in Weakness 1.

---

> > ### Author Response · Authors · 2024-11-22
> >
> > **Ablations on $\alpha, Q, r$**
> >
> > Appendix D studies the effect of hyperparameters $\alpha$ and $Q$ in an extended ablation study on MultiMNIST. Section 5.4 on Continuous Pareto Front expansion and the associated Figure 4 also ablate $\alpha$. Finally, Figures 8 and 9 further explore the relationship between $alpha$ and $Q$. Moreover, following the suggestion of the reviewer, we have included an ablation on the rank. The ablation is performed on Cityscapes and we consider ranks $r\in\{1,2,4,8\}$ and $M\in\{3,5\}$.The results are presented in Table 3 and show that our method is robust to the choice or rank.
> >
> >
> >
> >
> > [1] Mixture-of-Subspaces in Low-Rank Adaptation, EMNLP 2024.
> >
> > [2] Loraretriever: Input-aware lora retrieval and composition for mixed tasks in the wild, ACL 2024.
> >
> > [3] HydraLoRA: An Asymmetric LoRA Architecture for Efficient Fine-Tuning, NeurIPS 2024.
> >
> > [4] Mixture-of-LoRAs: An Efficient Multitask Tuning for Large Language Models, COLING 2024.
> >
> > [5] Pareto Manifold Learning: Tackling multiple tasks via ensembles of single-task models. ICML 2023

---

> > > ### Author Response · Authors · 2024-12-01
> > >
> > > Dear Reviewer vwtQ,
> > >
> > > As the rebuttal phase is nearing its conclusion, we would like to ask if our responses have addressed your concerns. If so, we would greatly appreciate it if you could reconsider your score.
> > >
> > > Kind regards,
> > >
> > > the Authors

---

### Meta-Review · Area_Chair_GhgZ · 2024-12-17

**Metareview:**

The contribution of this work lies in utilizing PFL for multi-task learning, demonstrating significant memory efficiency and scalability compared to prior methods. The proposed deterministic sampling strategy for PFL introduces a well-motivated annealing mechanism that improves convergence speed and stability. Extensive experiments across benchmarks substantiate its effectiveness, despite occasional parity with existing multi-task learning methods. While reviewers have concerns on figure/table clarity and some implementation details, the authors sufficiently address their concerns. Weighing all the factors, I believe this paper is marginally above the threshold for acceptance.

**Additional Comments On Reviewer Discussion:**

N/A

---

### Decision · Program_Chairs · 2025-01-22

Accept (Poster)